# Aeolus wind lidar observations of the 2019/2020 Quasi-Biennial Oscillation disruption with comparison to radiosondes and reanalysis

Timothy P. Banyard[1], Corwin J. Wright[1], Scott M. Osprey[2, 3], Neil P. Hindley[1], Gemma Halloran[4], Lawrence Coy[5, 6], Paul A. Newman[5], Neal Butchart[7], Martina Bramberger[8], and M. Joan Alexander[8]

[1]Centre for Space, Atmospheric and Oceanic Science, University of Bath, Bath, UK
[2]National Centre for Atmospheric Science, Oxford, UK
[3]Department of Physics, University of Oxford, Oxford, UK
[4]Met Office, Exeter, UK
[5]NASA Goddard Space Flight Center Greenbelt, Maryland, USA
[6]SSAI, Lanham, Maryland, USA
[7]Met Office Hadley Centre, Reading, UK
[8]NorthWest Research Associates, Boulder, Colorado, USA

**Correspondence:** Timothy P. Banyard (tpb38@bath.ac.uk)

**Abstract.** The quasi-biennial oscillation (QBO) was unexpectedly disrupted for only the second time in the historical record during the 2019/2020 boreal winter. As the dominant mode of atmospheric variability in the tropical stratosphere, and a significant source of seasonal predictability globally, understanding the drivers behind this unusual behaviour is very important. Here, novel data from Aeolus, the first Doppler wind lidar (DWL) in space, is used to observe the 2019/2020 QBO disruption.

Aeolus is the first satellite able to observe winds at high resolution on a global scale, and is therefore a uniquely capable platform for studying the evolution of the disruption and the broader circulation changes triggered by it. This study therefore contains the first direct wind observations of the QBO from space, and exploits measurements from a special Aeolus scanning mode, implemented to observe this disruption as it happened. Aeolus observes easterly winds of up to 20 m s$^{-1}$ in the core of the disruption jet during July 2020. By co-locating with radiosonde measurements from Singapore and ERA5 reanalysis,

comparisons of the observed wind structures in the tropical stratosphere are produced; showing differences in equatorial wave activity during the disruption period. Local zonal wind biases are found in both Aeolus and ERA5 at the tropopause, and the average Aeolus-ERA5 Rayleigh horizontal line-of-sight random error is found to be 7.58 m s$^{-1}$. The onset of the QBO disruption easterly jet occurs 5 days earlier in Aeolus observations compared with the reanalysis. This discrepancy is linked to Kelvin wave variance that is 3 to 6 m$^2$ s$^{-2}$ higher in Aeolus compared with ERA5, centered on regions of maximum vertical

wind shear in the tropical tropopause layer that are up to twice as sharp. An investigation into differences in the equivalent depth of the most dominant Kelvin waves suggests that slower, shorter vertical wavelength waves break more readily in Aeolus observations compared with reanalysis. This analysis highlights how Aeolus and future DWL satellites can deepen our understanding of the QBO, its disruptions, and the tropical upper-troposphere lower-stratosphere region more generally.

# 1  Introduction

The Quasi-Biennial Oscillation (QBO) is a regular cycle of alternating, downward propagating westerly and easterly winds which dominates the behaviour of the tropical stratosphere (Wallace, 1973). Observed continuously since 1953 using radiosonde measurements (Naujokat, 1986), the QBO has a relatively predictable period of $28 \pm 4$ months with a typical maximum amplitude between 20 and 30 m s$^{-1}$ (Bushell et al., 2020). In addition to its impact on the tropical tropopause, and therefore convection and related phenomena such as the Madden-Julian Oscillation (Feng and Lin, 2019; Lim et al., 2019), the QBO has also been shown to modulate the atmospheric circulation in the extratropical regions, and is an important source of predictability globally (Scaife et al., 2014). Furthermore, given that it is primarily a wind-based phenomenon, any new methods of measuring wind in this region of the atmosphere present a good opportunity to study the QBO from a completely different perspective.

Launched in September 2018, Aeolus is the first wind lidar in space. It is capable of measuring winds at high vertical resolution almost globally in the lowermost 30 km of the atmosphere, and thus theoretically the lower portion of the QBO. Since the QBO is highly technically challenging to model, wind observations are vital to understand it fully (Smith et al., 2022). Aeolus therefore presents a novel opportunity to measure the QBO directly and in its full zonal extent for the first time. To realise this potential, a special campaign involving a change to the satellite's onboard settings was initiated, raising the highest measuring altitude to observe a greater depth of the QBO (ESA, 2020a).

Despite the remarkable consistency of the QBO since 1953, since 2016 there have been two major and unprecedented disruptions to its expected evolution. The first, beginning in late 2015, was almost certainly caused by extratropical Rossby wave propagation during the Northern Hemisphere winter; and has already been studied extensively (e.g. Osprey et al., 2016; Newman et al., 2016; Coy et al., 2017; Barton and McCormack, 2017; Kang et al., 2020, and others). The second began in late 2019, and it has been proposed that anomalous Rossby wave activity from the Southern Hemisphere was partly responsible for this event (Kang and Chun, 2021). Irrespective of their causal mechanisms, neither disruption was well predicted by forecast models. Furthermore, the possibility of a link to anthropogenic climate change (Anstey et al., 2021) in addition to the high possible impact on future predictability, necessitates a more robust explanation for the dynamical processes which drive such phenomena.

This study explores the evolution of the 2019/2020 QBO disruption using measurements from Aeolus. It seeks to shed light on why forecast models struggled to predict the disruption, evaluates the role of Kelvin waves in its development, and thus investigates whether Aeolus and similar satellites could help to improve such predictions in the future. Section 2 outlines the methods involved in this study, with a description of the data used from Aeolus, Singapore radiosondes and ERA5 reanalysis. Section 3 shows the evolution of the disruption from multiple angles; including a validation of the data used, a look at the equatorial waves involved and an analysis of Kelvin wave equivalent depths using power spectra of the tropical winds. Section 4 offers explanations for the findings of this paper, a discussion of this study's limitations and a summary of the key points for future work. Section 5 concludes this paper with an outline of its main findings.

## 2 Data and Methods

In this study, the 2019/2020 QBO disruption is observed using data from the Aeolus wind lidar satellite, a tropical radiosonde station in Singapore, and ERA5 reanalysis. Together, these sources provide complimentary measurements of the evolution of the disruption from both global and local perspectives. In this section, each of the datasets is introduced, as well as the procedures followed to provide an appropriate comparison between them. Also explored are the methods used to best highlight the QBO's important characteristics, and the details of the analytical techniques used to deconstruct the causes and effects of the disruption itself.

### 2.1 Aeolus

Aeolus is a satellite launched in August 2018 carrying the first space-borne wind lidar instrument, called the Atmospheric Laser Doppler Instrument (ALADIN) (ESA, 1989, 2008; Chanin et al., 1989; Stoffelen et al., 2005; Reitebuch, 2012). As described by Banyard et al. (2021), its mission is to provide high vertical resolution profiles of wind, aerosol and cloud along its orbital path, with near global coverage. Observations are made by measuring backscattering from atmospheric molecules (Rayleigh scattering), and aerosol and hydrometeors (Mie scattering) along the laser's line-of-sight (LOS). Typically, the instrument measures from the surface up to around 20 km, however special configurations have enabled measurements to be made up to 30 km in specific regions and at certain times (ESA, 2020b; Legras et al., 2022).

The satellite's polar orbit is sun-synchronous with 15.6 orbits each day and a repeat cycle of 7 days. For the duration of the observing period in this study there is a close overpass to the site of the Singapore radiosonde station between 22:55 and 23:00 UTC every Wednesday. Aeolus' orbit has an inclination of 96.97°, resulting in a near-meridional orbital path at the equator, and it flies at a mean altitude of 320 km, with an ascending-node local equator-crossing time of 18:00.

To enable the instrument to provide an observation of the horizontal wind, both laser and telescope are directed at 35° off-nadir, perpendicular to the direction of travel. Along the laser's LOS, a single wind component is measured, which is then converted into the horizontal line-of-sight (HLOS) wind speed $v_{HLOS}$ through the assumption that the vertical wind speed is small. The resulting $v_{HLOS}$ measurements are therefore near-zonal in the tropical lower stratosphere, which is particularly advantageous for observing the QBO.

A number of methods can be implemented to extract the true zonal component of the wind from the original HLOS product provided by Aeolus, as discussed by Krisch et al. (2022). Here, given that the analysis is constrained to the equatorial region, the raw HLOS winds can be used as a good approximation for the zonal wind. Furthermore, in comparisons with radiosonde and reanalysis data, all non-Aeolus data is reprojected onto the Aeolus HLOS direction, thus eliminating any potential biases caused by geometry. Therefore, whenever the 'zonal wind' is referred to in this study, it is technically the 'near-zonal wind' which is being analysed; however, for simplicity, the phrase 'zonal wind' will be used. Equations describing Aeolus' measurement geometry can be found in Banyard et al. (2021), and a more detailed description and exploration of its limitations is in Krisch et al. (2022).

Raw Aeolus observations are processed to produce measurements of the HLOS wind speed and placed in 24 vertical "range bins", each with thicknesses between 250 and 2,000 m, which can be modified by the satellite operators. Numerical Weather Prediction (NWP) requirements are such that the vertical resolution of Aeolus wind data is around 1 km in the Upper-Troposphere Lower-Stratosphere (UTLS) region, with this tending towards 2 km in the lower stratosphere. This inevitably means the data is relatively coarse in the region of the QBO, and therefore at the altitude of the 2019/2020 disruption. To better observe the evolution of the QBO following this event, a special range-bin setting (RBS) was implemented onboard the spacecraft (see Fig. 1) beginning in June 2020. This raised the maximum altitude of each wind profile by 5000 m to 25.5 km in the latitude range 10°S to 10°N each Wednesday. Bins lower down were also modified to maintain an appropriate bin spacing throughout the atmospheric profile. Since the QBO varies over a long temporal scale, the implementation of this RBS once per week is sufficient to provide suitable information about the QBO's evolution in the lower stratosphere.

In this study, the most recently available Level 2B (L2B) product as of February 2023 is used for each time, ranging progressively from processing Baseline 11 to Baseline 14, which includes some reprocessed data for certain periods during the mission. The improvements made for the later baselines include a new parameterisation to simplify the Rayleigh Brillouin Calibration (RBC) table which is used in processing (Baseline 13), and the removal of a seasonal ascending/descending node bias (Baseline 14). Only Rayleigh winds are considered here in order to both maximise data coverage and simplify the processing of the analysis. Future studies could include Mie winds as well, especially since these may provide useful additional skill in the tropical upper-troposphere at cloud-top level.

Quality controls are applied, including (i) a restriction to only use winds in clear-sky conditions, particularly since Rayleigh-cloudy data was unreliable prior to the aforementioned Baseline 13 improvement, and (ii) a cut off of 8 m s$^{-1}$ on the random error for each data point. Recent literature shows systematic biases of < 1 m s$^{-1}$ for the Rayleigh wind L2B product (Abdalla et al., 2020), suggesting the dominant source of error is the random error of the instrument. There are two main sources of random error for the ALADIN instrument, which act to decrease the signal-to-noise ratio (SNR) and/or broaden the spectral width of the backscattered signal. As discussed by Reitebuch et al. (2020), of greatest importance is the signal photon shot noise, which arises from fluctuations in the number of arriving photons from the expected average. Secondarily there is also the influence of unwanted electronic noise, which is caused by other electronic components in the detection chain of the instrument. Importantly for future Doppler wind lidar (DWL) satellites, Aeolus exhibited a suppressed Rayleigh signal when compared with its designed capabilities, suggesting that low SNR may become less of an issue for future instruments.

In addition, Lux et al. (2022) showed that the random error gradually increased over the time duration of the FM-B laser portion of the mission, in line with a gradual decrease in the return energy from each laser pulse from the ALADIN instrument. Although for some purposes this change in random error poses less of an issue, such as case studies covering less than a day, and studies involving calculations of wind perturbations (e.g. Banyard et al. (2021)), when looking at raw winds in the UTLS this error component could pose a bigger problem. For this study, the problem of random error is alleviated somewhat where a daily zonal-mean wind is calculated, due to the large number of data points that are included. The time-varying random error will however have a greater impact on the results from the validation exercise in Sect. 3.2, with comparisons to radiosonde and reanalysis data. Although these results cover a shorter time frame, the ALADIN instrument did still experience some minor

fluctuations in energy output during this period. Further information about the Aeolus data products and retrieval algorithms is given by Tan et al. (2008), and is expanded upon in the Algorithm Theoretical Basis Documents (ATBD) for L1B (Reitebuch et al., 2018) and L2B (Rennie et al., 2020) data.

## 2.2 Singapore Radiosonde

The tropical radiosonde station at Singapore (1°N, 104°E) produces twice-daily high-resolution meteorological soundings which are stored as part of the Integrated Global Radiosonde Archive (Durre et al., 2018). Data from this station is used partly due to the good continuity and longevity of its record; indeed the Singapore radiosonde is often used as a proxy for the QBO's historical progression and is considered the 'gold standard' in QBO research. For the majority of the time period covered in this analysis, this station used the Vaisala VRS41-SG radiosonde (Vaisala, 2014) and Vaisala DigiCORA III sounding system to receive and process wind information, giving values for the wind speed and direction with an uncertainty of 0.15 m s$^{-1}$ and 2° respectively. Each radiosonde reaches a typical altitude of around 30 to 35 km and provides data at a mean vertical resolution of 250 m. Due to the weekly recurrence of a special setting onboard Aeolus for measuring the QBO, designed to coincide with one of the closest overpasses with an average minimum proximity to the radiosonde station of ∼80 km, only the 00:00 UTC sounding from Singapore on a Thursday is used. Realistic comparisons with Aeolus winds are obtained by projecting the radiosonde winds onto the same direction as the Aeolus measurement geometry. Although the geographical discrepancy between the two datasets is not negligible, its effect on the results are not deemed significant given the context of the QBO, which is a large-scale atmospheric phenomenon. Data used in this study covers the time period from mid-2019 to late-2021.

## 2.3 ERA5

ERA5 is an atmospheric reanalysis data set provided by the European Centre for Medium-Range Weather Forecasts (ECMWF). It is a combination of an earth-system model with assimilated observations which supplies an historical archive of the atmospheric state globally (Hersbach et al., 2020). ERA5 has a spatial resolution of 0.25°×0.25° (∼31 km), 137 vertical levels from the surface up to 0.1 hPa, and the data-set used here has a temporal resolution of three hours. For the purposes of this study, ERA5 data has been projected onto the Aeolus HLOS for each data point to create a matching dataset which simulates Aeolus orbiting through an ERA5-like atmosphere. This allows the same analysis to be conducted on both Aeolus and ERA5 data to give appropriate comparisons between the two. Although Aeolus data is not assimilated into ERA5, observations from the Singapore radiosonde are, which is an important factor to consider during the subsequent comparisons. The formula for calculating synthetic ERA5 HLOS winds from the $u$ and $v$ horizontal cartesian wind components is given below:

$$v_{\text{HLOS}} = -u\sin(\Psi) - v\cos(\Psi),$$ (1)

where $\Psi$ is the bearing of the satellite track at each data point.

Although there are no significant biases that affect the conclusions of this study, there are some that may be relevant to the analysis that is carried out; most of which are discussed by Shepherd et al. (2018). Most notable is a global-mean temperature

bias in the underlying model, (Integrated Forecasting System) IFS cycle 41r2, used to produce the reanalysis. This resolution dependent bias arises from the energy budget of upward-propagating gravity waves and produces a persistent cold offset of up to 0.5 K. The bias maximises in severity at around 70 hPa, and is larger than was present in the model version used for ERA-Interim. Nonetheless, Tegtmeier et al. (2020) found that ERA5 showed the most realistic tropopause temperatures when compared with other atmospheric reanalysis products, justifying its use for this study. Secondly, radiosonde observations have been found to correct lower stratospheric biases less effectively in ERA5 compared with ERA-Interim. This is mainly due to changes in the mesoscale spectrum and larger specified radiosonde errors. This bias appears to have improved in recent years however, since the assimilated satellite observations, in particular from Global Positioning System - Radio Occultation (GPS-RO), have increased. Finally, there are biases in the upper portion of the QBO which is generated by the model through wave, mean-flow interaction from tropical radiosonde measurements in the lower stratosphere. An example of this is an unrealistically strong westerly mesospheric jet which was generated in connection with the Semi-Annual Oscillation (SAO) during March 2016. Since this study focuses on the UTLS region of the atmosphere, this particular bias is not expected to impact the results here. These issues are however important to consider in any comparison of ERA5 to observations, as is done in this study.

In addition to the above recognised biases, ERA5 is also known to exhibit uncertainties in the lower stratosphere. Kawatani et al. (2016) showed that atmospheric reanalyses generally show greater uncertainty in tropical zonal winds at the tropopause, with a small improvement at around 70 hPa and then increasing uncertainty with altitude into the upper stratosphere. Although that study does not consider ERA5, many of the issues it investigates are likely to be common to all atmospheric reanalyses. Healy et al. (2020) demonstrates root-mean-squared (RMS) zonal wind differences between Global Navigation Satellite System - Radio Occultation (GNSS-RO) and ERA5 of up to 5 m s$^{-1}$ at 30 hPa, and the ECMWF's own uncertainty estimations from the Ensemble of Data Assimilations (EDA) system (Hersbach et al., 2020) show random errors in excess of 3 m s$^{-1}$ above 200 hPa in ERA5.

## 3 Results

### 3.1 Aeolus observations of the QBO disruption

Both QBO disruptions were characterised by the stalling of the normal descent of the zonal wind bands in the tropical stratosphere and the formation of a second anomalous easterly layer within the lower stratospheric westerly QBO phase. Figure 1 portrays the evolution of the 2019/2020 QBO disruption, and the subsequent QBO cycle, as observed by Aeolus. The initial disruption at the end of 2019 is followed in late 2020 and 2021 by the resumption of the descent of the active westerly phase of the QBO, with the next QBO cycle appearing towards the end of the time series.

Only the lowermost portion of the QBO is observed by Aeolus due to the constraints imposed by the RBS, but nonetheless, the beginning of the wind reversal to easterlies can be seen clearly during December 2019 at 22 km. This is preceded by a narrow, ascending region of suppressed westerlies, emanating from near the tropopause during August-September 2019, which is marked by the dashed blue arrow.

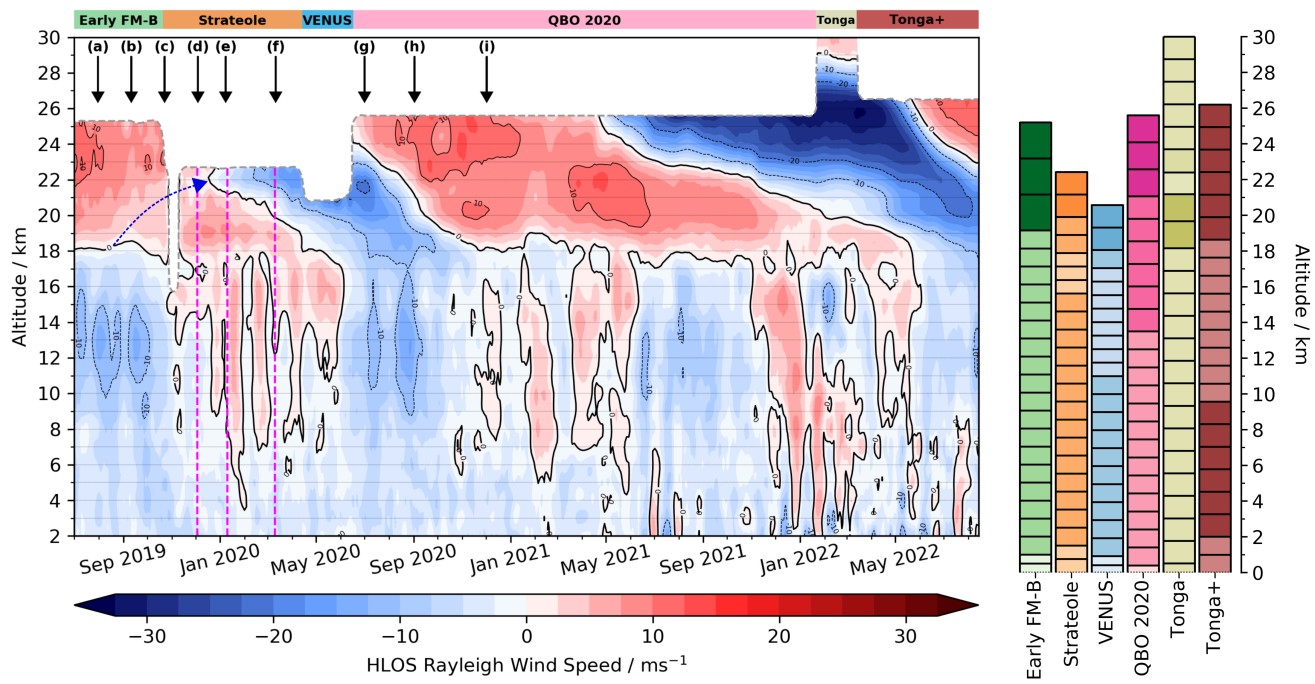

**Figure 1.** Timeseries of daily zonal-mean HLOS wind from Aeolus between 5°S to 5°N. The grey dashed line signifies the altitude of the highest range-bin observed at that time. This altitude varies according to a series of successive RBS changes, which are illustrated with a bar plot on the right hand side of the figure, with lighter shades denoting shallower bins. The duration of each respective RBS for this latitudinal range is shown by coloured bars above the time series. The 9 black arrows shown just underneath correspond to the dates of the profile comparisons shown in figure 3(a-i). The dashed blue arrow marks the ascending region of suppressed westerlies which precedes the disruption. The 3 dashed magenta lines during the period of the disruption correspond to the cross-sectional snapshots in Fig. 7. A 7-day boxcar filter is used to fill the gaps above 20 km following the QBO 2020 RBS change; and where there are data gaps elsewhere, these are filled using a broader 20-day boxcar filter.

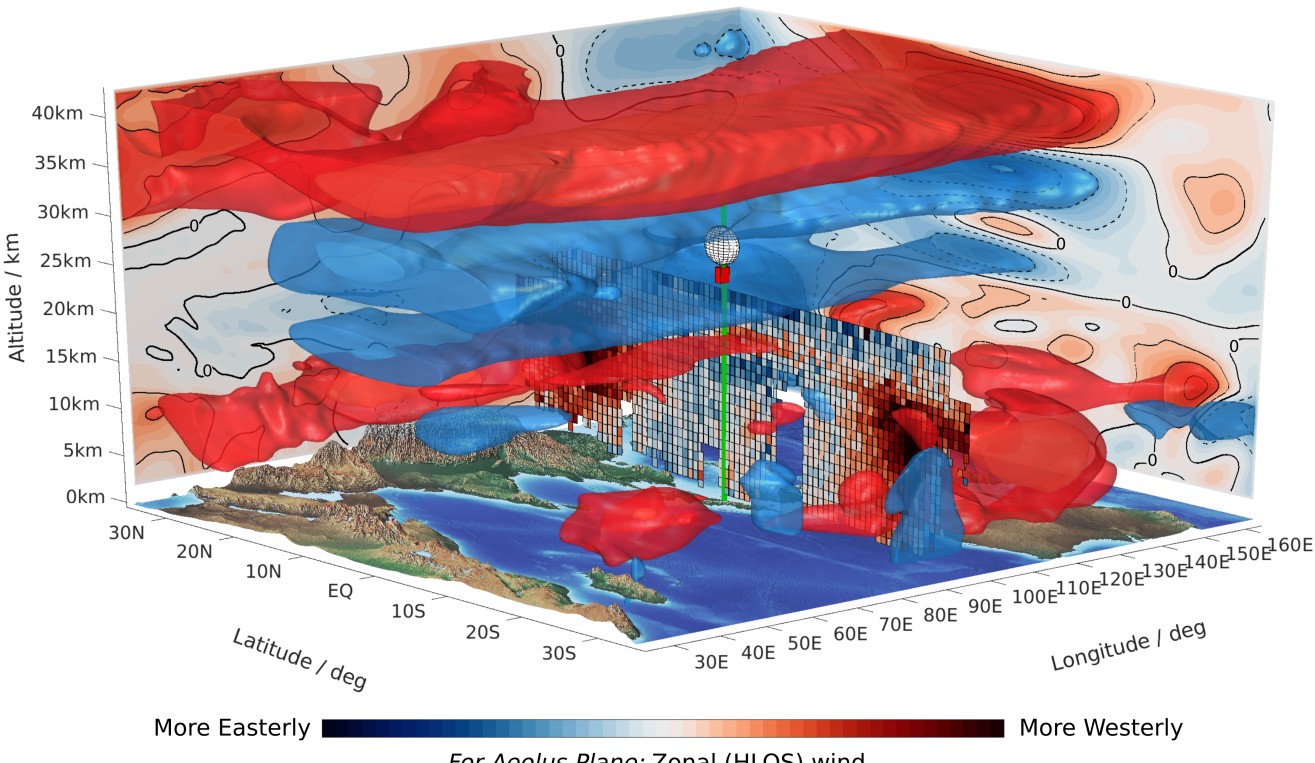

ERA5 Daily Mean: 2020-04-01  |  Aeolus Orbit: 21:50:23  |  Singapore Radiosonde Ascent: 2020-04-02 00z

More Easterly ▬▬▬ More Westerly

*For Aeolus Plane:* Zonal (HLOS) wind
*For ERA5 2D and 3D contours:* Zonal wind anomaly (1991-2020 Daily Climatology)

**Figure 2.** Illustration of the three datasets used in this study and the manner in which they relate to each other. ERA5 zonal wind anomalies are shown using coloured contours, with 3D contours at -22, -11, 11 and 22 m s$^{-1}$ to highlight the important features of the QBO disruption. Along-track profiles of Aeolus data and the launch trajectory of the Singapore radiosonde are also shown, with the exact location of each data point used for both data sets. This figure is intended to give a qualitative perspective which helps the reader understand the broader context behind the data which gives the results in this paper.

Following the wind reversal, the region of disruption easterlies expands vertically with time, with the lower wind shear zone progressing downwards in a QBO-like manner. In July 2020 this easterly jet reaches a maximum magnitude of 20 m s$^{-1}$ at 22 km, although this measurement is hindered by the RBS at the time. The new RBS then reveal the now downward-propagating upper wind shear zone in mid-2020, as westerly winds quickly follow the disruptive easterlies. Later in 2021, the highest range-bins begin to show the subsequent easterly QBO phase, which progresses uninterrupted through to nearly the end of the data period. During 2022, new RBS to measure the stratospheric effects of the volcanic plume from the 15 January eruption of the Hunga-Tonga volcano are introduced. These reveal the next westerly phase of the QBO, which descends from the top range-bin of 30 km to around 23 km by the end of the period.

Throughout much of the time-series, a series of westerly wind pulses can be seen in the middle- and upper-troposphere, in the midst of more persistent easterly winds which dominate the tropical troposphere. The temporal frequency of these pulses suggests the occurrence of significant tropical wave activity, likely related to Kelvin waves, particularly during the evolution of the disruption following its triggering in late-2019. Further analysis on the propagation of these waves along the equator is carried out later in this study, and three representative snapshots of their vertical structure, corresponding to the times marked by the magenta dashed lines, will be considered in section 3.3.

These pulses of westerly winds exhibit a seasonality which can likely be explained by phenomena called westerly wind bursts (WWBs), which are thought to play an important role in the initiation of El Niño events. Driven at least in part by atmospheric Kelvin waves, Pacific-basin WWBs induce anomalous ocean surface wind stresses and initiate oceanic Kelvin wave propagation through the equatorial Pacific, deepening the thermocline and accelerating the onset of El Niño conditions (Tan et al., 2020). As noted by Seiki and Takayabu (2007), WWBs appear to be more common between November and April, matching the stronger westerly winds seen in Fig. 1 during the boreal winters of 2019/2020, 2020/2021 and 2021/2022. Some literature has discussed the presence of a so-called 'spring predictability barrier' which causes El Niño forecasts to rapidly increase in skill with decreasing lead-time throughout the boreal spring period (Latif et al., 1998; Zheng and Zhu, 2010; Chen et al., 2020). Its timing is often linked with the occurrence of such WWBs due to the oceanic processes that are triggered. More accurate measurements of WWBs provided by space-borne wind lidars such as Aeolus could therefore prove useful in combatting predictability issues such as this.

The fine vertical resolution of Aeolus data in the UTLS allows the behaviour of winds in the vicinity of the tropopause to be observed in detail. Due to the strong turbulence and vertical winds that distinguish the troposphere from the stratosphere, the QBO does not extend below the tropopause, although some previous studies have observed a weak oscillation comparable to the QBO at the tropopause itself (Angell and Korshover, 1964; Reid and Gage, 1985). The tropopause is a boundary most often defined by the altitude at which the vertical gradient in temperature reverses, commonly known as the cold point tropopause (CPT) (Highwood and Hoskins, 1998). Since the QBO is a wind phenomena however, it is useful to understand the lower limit of its downward propagation from a wind based perspective. Using GNSS-RO data, Tegtmeier et al. (2020) showed that the amplitude of the QBO at the CPT varies with longitude, and there is uncertainty about both the spatial and temporal variability of the QBO's vertical extent. In the timeseries in Fig. 1, the lower limit of the QBO in a zonal-mean framework remains steady between around 17 and 18 km, around a kilometre above the typical tropopause altitude between 5°N and 5°S.

Since the tropical tropopause layer (TTL) is a region where important transfers of energy, chemical species and water vapour occur, the addition of high vertical resolution wind measurements from Aeolus and similar wind lidars is desirable, particularly for assimilation into atmospheric reanalyses and NWP models. Currently, radiosonde analyses form much of our understanding about TTL processes, particularly since these pre-date the satellite era and give a longer historical record. Later in this study, 9 radiosonde profiles from Singapore are compared with ERA5 and Aeolus; the timings of these during the QBO disruption are denoted with arrows which are marked (a) to (i) in Fig. 1.

To the right of the timeseries in Fig. 1 are six stacked bar charts showing the different RBS that are active throughout the measurement period, illustrating the importance of the QBO 2020 RBS for observing the QBO disruption and its subsequent evolution, with an increase in the top altitude of $\sim$5 km. One of the unintended consequences of the Tonga RBS change in January 2022 is observations of the QBO up to 30 km using Aeolus for the first time, allowing the initial signs of westerly winds associated with the next QBO phase to be observed above $\sim$28 km. Despite the decreased SNR at these heights, the large number of data points (up to $\sim$200) being used for a zonal-mean daily wind value between 5°N and 5°S means that this is likely to still be a trustworthy result. Since much of the dynamical forcing associated with the QBO takes place at such altitudes, wind measurements here by future DWL satellites could prove very useful, both for deepening scientific understanding and improving model predictions of the QBO.

## 3.2 Validation against reanalysis and radiosondes

In order to validate the data from Aeolus as the disruption evolves, co-located ERA5 reanalysis and high altitude radiosonde launches from Singapore are used. Fig. 2 illustrates these three datasets in relation to each other, showing the location of the radiosonde launch site and the closest intersecting Aeolus orbit, as well as the context of the ERA5 zonal wind anomaly at the time of the disruption itself. The back plane of the figure shows an equatorial cross-section of the zonal wind near 160°E, from ERA5, which matches up with the 3D contours plotted throughout the image.

Clear spatial and temporal limitations of the comparison between the datasets are seen. Notably, the radiosonde's location is offset from the Aeolus orbit, and barely drifts horizontally as it ascends (consistent with Seidel et al. (2011)); and only a single descending-node orbit is used since it is the closest to the radiosonde launch site. Additionally, as mentioned by Kawatani et al. (2016), it is possible that the radiosonde data strongly nudges the reanalysis winds towards its own. Conversely, this may lead to the higher variability of winds across different reanalyses in data sparse regions, such as over the Pacific ocean where there are no radiosonde launches, where each reanalysis will drift towards the model's climatology. Such a dependence of the reanalysis on radiosonde data is important to remember throughout this analysis.

In Fig. 2, the disruption easterly and underlying westerlies can be seen clearly in the ERA5 zonal wind anomaly and Aeolus overpass. Also notable is the widening in latitudinal extent of the QBO and disruption anomalies with increasing altitude, all the way up to 40 km, a property of the QBO which is observed in previous literature (Reed, 1965). The subtropical jet streams can also be observed in this figure, and are shown for completeness.

The perspective given by the along-track profiles of Aeolus also demonstrates the acute angle that the satellite's orbit follows relative to the meridian, which leads to the Aeolus HLOS winds being very representative of the zonal wind, as the ALADIN

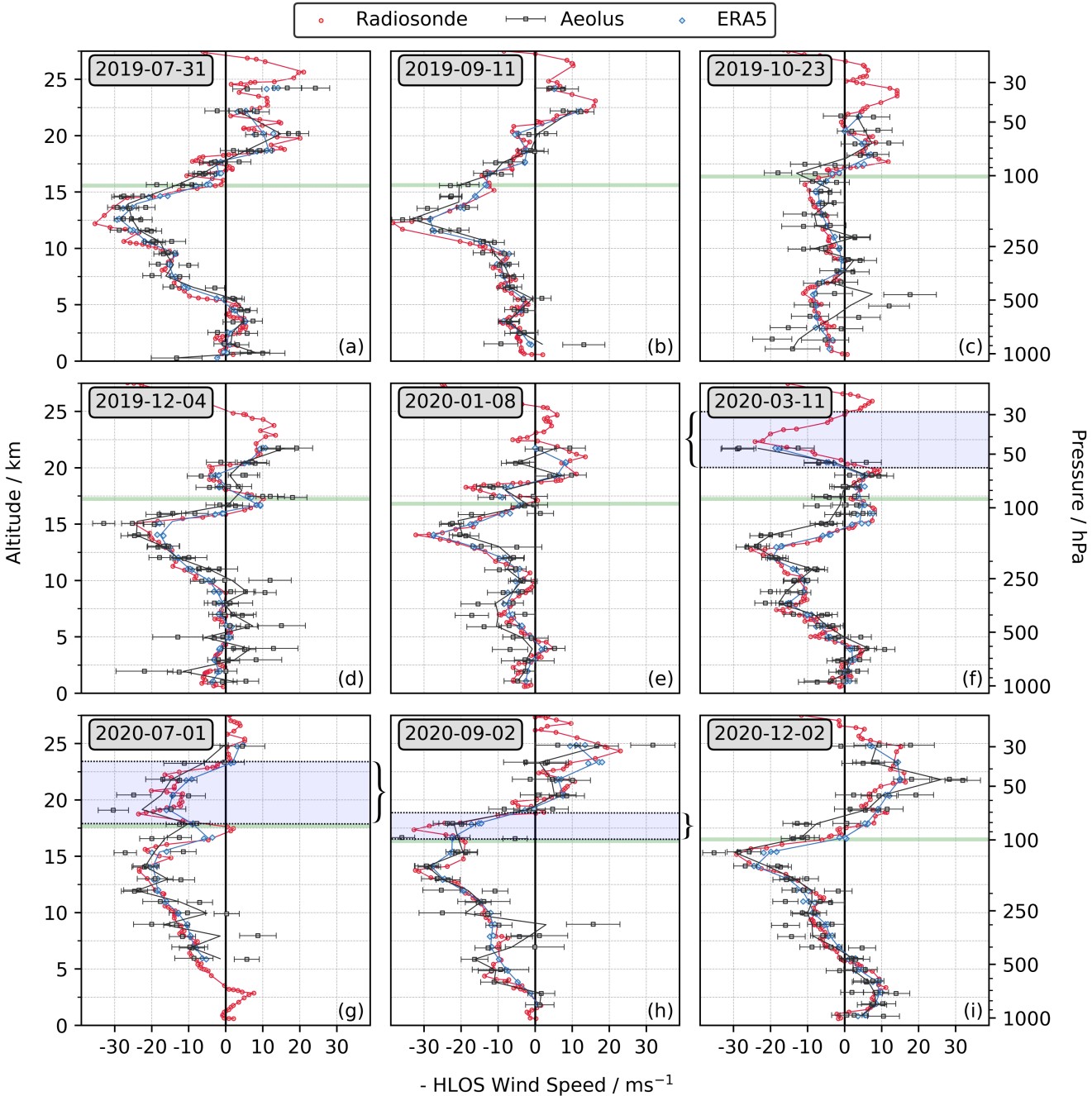

**Figure 3.** (a)-(i) Profiles of the HLOS wind from Aeolus (black), projected-HLOS wind from ERA5 (blue) and projected-HLOS wind from Singapore-launched radiosondes (red) for a progression of time-steps during the QBO disruption. Due to orbital geometry, wind values are sign-flipped, making easterly (westerly) winds appear negative (positive) as required by convention. Lines follow the average of profiles within the 150 km radius surrounding area at each range-bin height for Aeolus and ERA5. Error bars shown for Aeolus correspond to the HLOS error estimates given in the L2B files. The ERA5 tropopause for each day is shown by a horizontal green line, and the easterly disruption region above the tropopause is marked by the pale blue region bounded by curly brackets. The corresponding timestamp is shown in the top-left corner of each subplot.

laser is directed perpendicular to the direction of travel. On close inspection, the latitudinal limits of the VENUS equatorial RBS, 30°S to 30°N, which existed prior to the QBO 2020 RBS, can be seen. Both subtropical jets in the northern and southern hemispheres, present during April 2020, can also be clearly seen polewards of 20°S and 20°N. Gaps in the Aeolus data are also visible, particular where there are more convective clouds such as in the intertropical convergence zone (ITCZ). One limitation of using just the Rayleigh clear Aeolus HLOS winds is that cloudy returns are removed by quality control, so the winds within the ITCZ are likely to be represented less well in this data.

A set of nine Aeolus overpasses, each overlaid with the corresponding Singapore radiosonde profile and ERA5 data for the same time, is shown in Fig. 3. Only Aeolus profiles within 150 km of the Singapore launch site are used since the larger-scale dynamics should not change significantly across this area; this also accounts for drift in the radiosonde location as it ascends. Since reprocessed Aeolus data is being used, any global-scale systematic biases that arise from using only descending node data should be kept low (Weiler et al., 2021). However, as noted by ESA (2021), there may still be higher regional biases present which could be visible in data from a single radiosonde station. In order to match the zonal wind speed direction, all Aeolus winds are sign-flipped on the descending node to become -HLOS, i.e. negative HLOS. This is required because ALADIN's laser is directed perpendicularly to the right hand side of the satellite's orbital path. The error in the Singapore radiosonde measurements is considered small relative to the given HLOS error from Aeolus, whereas ERA5 is expected to exhibit more uncertainty in the wind speeds at these altitudes, particularly given the analysis of Kawatani et al. (2016) and Healy et al. (2020) as discussed in Sect. 2.3.

From (a-i), each set of profiles shows a representative snapshot throughout the disruption period. Dates are chosen to give a good representation of the entire disruption, from its initiation, to its evolution and after-effects, and only where good quality data is available from all datasets.

Good agreement between all three datasets is seen throughout, with the same persistent easterly winds which are seen dominating the tropical troposphere in Fig. 1, clearly shown. From Fig. 3b-e, the weakening of westerly winds which precedes the disruption is visible between 17 and 23 km. Then, in Fig. 3f, the disruption easterly itself becomes clearly discernible in all three datasets, with local easterly winds reaching 20 m s$^{-1}$ in mid-March around 22 km. This region has deepened and descended slightly by early-July (Fig. 3g) such that only a thin sliver of westerly winds can be seen in the radiosonde data around the altitude of the tropopause. By the end of 2020 (Fig. 3i), the westerly QBO has descended through the lower stratosphere, although the altitude of the zero-wind zone is around 2 km higher in Aeolus compared with the radiosonde and ERA5. A similar discrepancy can be seen during mid-March (Fig. 3f), with Aeolus observing winds around 14 km that are up to 15 m s$^{-1}$ more easterly than the comparison datasets. The absence of Aeolus winds below 5 km in Fig. 3g is most likely due to cloud in the lower troposphere, and the slightly higher random error throughout the atmospheric profile in this and the profile in Fig. 3h could be attributed to cloud contamination in the backscattered Rayleigh signal.

By combining all available profiles from each Wednesday during the time period, selected to correspond with the day the QBO 2020 RBS is active, Fig. 4 shows distributions of the differences between the 3 datasets for (a) Aeolus - Singapore Radiosonde, (b) Aeolus - ERA5 and (c) Singapore Radiosonde - ERA5. Since the radiosonde data is assimilated into ERA5, the significantly lower difference between these two datasets (Fig. 4c) is to be expected. Once again, good agreement between all

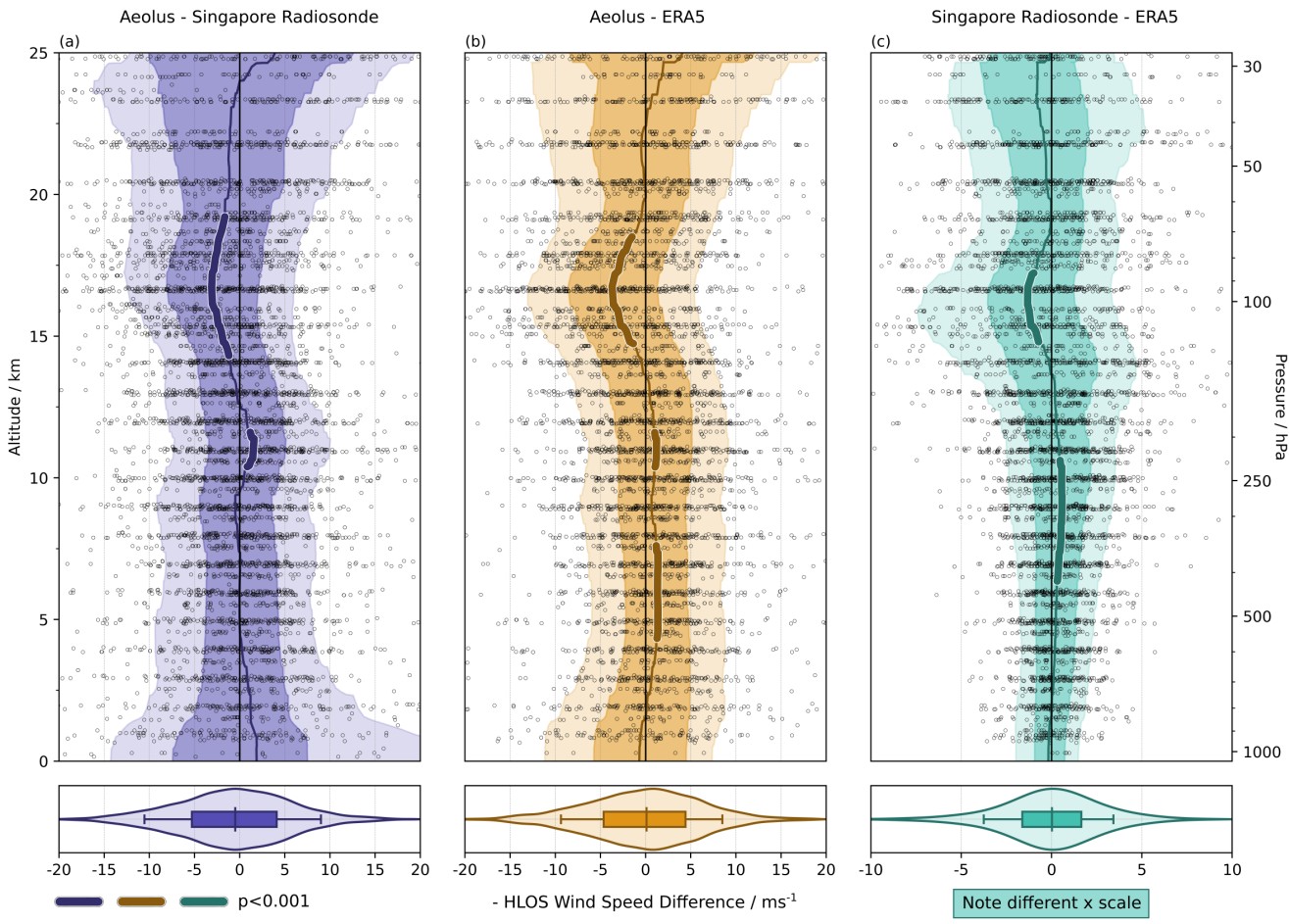

**Figure 4.** Vertical distribution of the point-by-point HLOS wind speed difference for (a) Aeolus-Singapore, (b) Aeolus-ERA5, (c) ERA5-Singapore. Lighter shades denote the area bounded by the 10th and 90th percentiles, darker shades bound the 25th and 75th percentiles and the median is marked by a solid coloured line. This line is thickened and outlined in white at altitudes where a Student's t-test returns p < 0.001. Dots show the individual point differences which make up the distributions shown. Below each profile is a violin plot showing the average distribution for all heights, with the median, and 10th, 25th, 75th and 90th percentiles indicated. The median differences and standard deviations are, for (a) -0.49 m s$^{-1}$ and 9.30 m s$^{-1}$, for (b) 0.09 m s$^{-1}$ and 7.58 m s$^{-1}$, and for (c) 0.02 m s$^{-1}$ and 3.31 m s$^{-1}$.

three datasets is seen, with median differences of -0.49 m s$^{-1}$ and 0.09 m s$^{-1}$ between Aeolus, and the Singapore Radiosonde and ERA5 respectively. The standard deviation across all heights is in broad agreement with existing literature (e.g. Rennie et al. (2021), Martin et al. (2021) and Lux et al. (2022)), although is slightly at the upper end of some estimates, with an average Aeolus-ERA5 difference of 7.58 m s$^{-1}$.

In all three comparisons, there is greater spread between the datasets at higher altitudes, which corresponds with Aeolus' reducing SNR with height and the well known issue of less representative reanalysis winds in the stratosphere compared with the troposphere (Baldwin and Gray, 2005; Kawatani et al., 2016; Sivan et al., 2021). The reason for the lower Aeolus SNR above the tropopause is simply the reduced Rayleigh backscattering caused by atmospheric density dropping exponentially with altitude (Reitebuch et al., 2020), and is a feature seen in pre-launch simulations of Aeolus winds (Rennie, 2018).

In order to identify regions where the differences are statistically significant, Student's t-tests have been applied at each altitude. The tests are constrained by requiring the threshold of $p < 0.001$ to be valid for 3 consecutive altitude bins, each with a depth of 2 km and overlapping at intervals of 200 m. Figure 4a shows that Aeolus winds are easterly-biased relative to radiosonde measurements between 14 and 19 km, reaching a maximal bias of around -3 m s$^{-1}$ near the tropopause itself, which is where the greatest differences tend to be for all three dataset comparisons. The bias becomes positive over the height range 10 to 12 km, but is largely negligible below this height. This height dependence of the Aeolus bias shows a comparable morphology to validation studies by Abril-Gago et al. (2023) and Ratynski et al. (2023), the latter showing a particularly similar systematic bias of around -2 m s$^{-1}$ between 13 and 20 km on descending node overpasses. Given the high vertical wind shear at these altitudes, as noted by Houchi et al. (2010), it is likely that the apparent dipole in this bias is related to local wind shear effects. On the one hand, Aeolus, through its long horizontal accumulation of measurements which form each wind profile, has the capacity to capture localised wind shear that the radiosonde might miss. On the other hand, radiosondes provide a better vertical resolution and can accurately capture regions of high wind shear, but only at the location where the radiosonde is taking measurements. Throughout the main depth of the atmospheric profile, the standard deviation of the difference between Aeolus and Singapore radiosonde winds remains largely constant at around 8 to 9 m s$^{-1}$.

    The most notable feature which stands out for both observing instruments relative to the reanalysis (Fig. 4b, c), is again a significant easterly wind difference around the tropopause, with both Aeolus and the Singapore radiosonde profiles exhibiting a pronounced, yet relatively shallow deviation centered around 17 km. It is known from previous work that atmospheric reanalyses have historically shown a peak in bias around the tropopause. For example, the presence of a CPT warm bias in NCEP-NCAR (R1) is discussed by Tegtmeier et al. (2020) as being primarily due to the vertical resolution of the model. Notably, in the tropics this is related to Kelvin wave activity, and likely results from the use of poorly resolved satellite temperature retrievals, as explored by Fujiwara et al. (2017). The most likely cause of temperature biases at the tropopause is the smoothing of sharp vertical temperature gradients in the data assimilation system, with studies such as Flannaghan and Fueglistaler (2011, 2014) identifying a strong sensitivity in atmospheric reanalyses to mixing from shear-flow instabilities in connection with Kelvin waves. Wind data from Aeolus has been shown to contribute the highest changes in NWP skill around the tropical tropopause height (Rennie et al., 2021; Martin et al., 2022), so it is suggested that this particular wind bias in the observing instruments seen in Fig. 4 is a consequence of the same issue related to vertical resolution and the failure

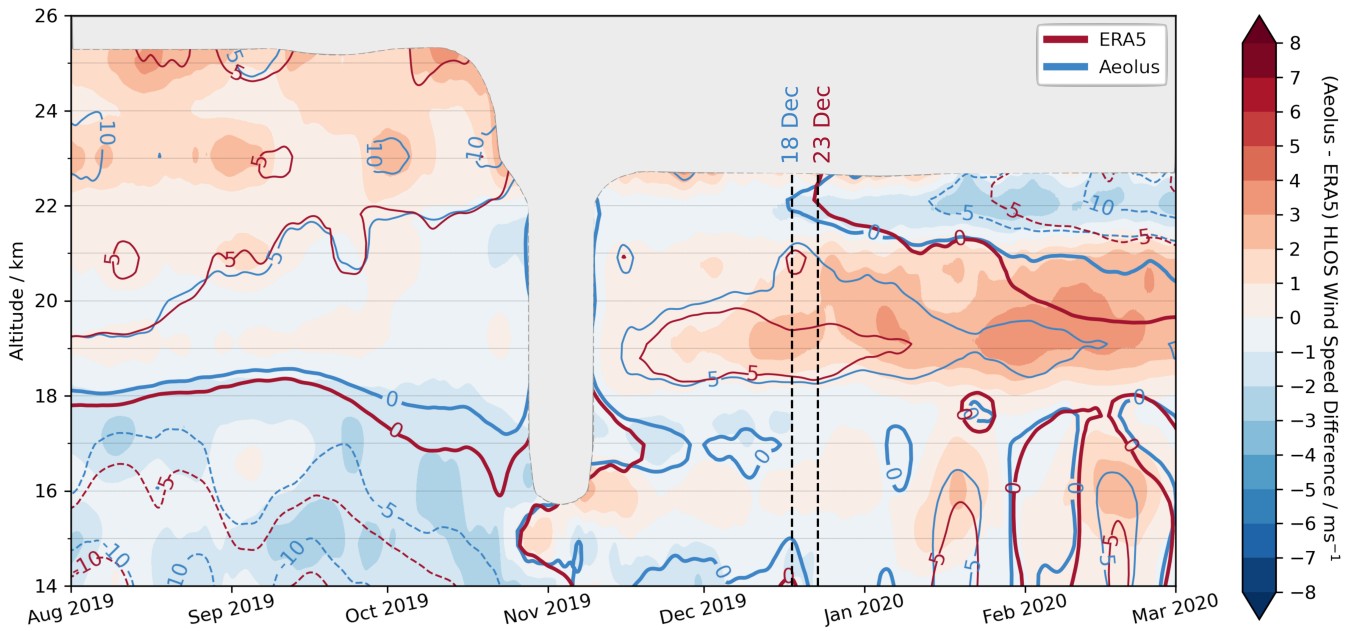

**Figure 5.** Timeseries of daily zonal-mean HLOS wind between 5°S to 5°N as in Fig. 1, but using coloured contours to show both Aeolus (blue) and ERA5 (red), and for the period August 2019 - March 2020. The difference between Aeolus and ERA5 is shown as filled contours in the background. The dashed black vertical lines mark the time of the onset of easterly winds at 22 km for Aeolus and ERA5, as 18 December 2019 and 23 December 2019 respectively.

of atmospheric reanalyses to accurately capture mixing due to Kelvin waves, as outlined above. In the context of the QBO disruption, which appears to have propagated upwards from the TTL region, the addition of Aeolus information both on its own and assimilated into future reanalyses is therefore likely to be very useful.

Another difference between Aeolus and ERA5 can be observed by comparing the timeseries of the QBO disruption in the zonal-mean HLOS wind for both datasets simultaneously. Figure 5 shows that there is great similarity between the two datasets when a direct comparison is made by projecting ERA5 onto the Aeolus HLOS winds as described in Sect. 2.3. However, there are still some notable differences. In particular, the onset of the easterly winds at 22 km which characterises the disruption itself is delayed in the reanalysis by 5 days with respect to Aeolus. This lag appears to persist in time and is seen again with the onset of the -5 m s$^{-1}$ isotach during January 2020.

When examining each timeseries independently, the reason behind the inconsistency in the disruption onset time initially remains unclear. The inclusion of both ascending and descending node measurements in each daily wind calculation eliminates any point-by-point systematic biases in either dataset as potential causes. However, when overlaying the difference between the two datasets onto the combined timeseries, it is possible to see some features which might explain the above discrepancy.

In general, Aeolus zonal-mean HLOS winds are greater in magnitude than in ERA5, which leads to a higher vertical wind
shear around 21 km. Consequently, Aeolus observes more easterly winds than ERA5 within the disruption easterly jet. This
alone however does not explain the time offset for the zero-wind isotach; for a period of time leading up to the wind reversal,
Aeolus also has more easterly winds. Furthermore, the dipole pattern across the altitude of highest wind shear is already
established before the wind shear itself develops. As the negative vertical wind shear into the disruption easterly steepens with
time, ERA5 continues to lag behind.

Two main reasons for this lag are therefore proposed. First, is the possibility that the reanalysis has a tendency to revert
to its model climatology, which hampers its ability to accurately track the progression of the disruption as measured using
observations, including Aeolus. By definition, the disruption itself is an anomalous feature in the tropical lower stratosphere
relative to climatology. The zonally inhomogeneous sampling of wind measurements assimilated into the reanalysis model
means that data from certain locations may bias the zonal-mean wind in ERA5. Conversely, Aeolus is measuring across all
longitudes with a near-homogeneous repeating orbital pattern, so the zonal-mean should be less affected by regional wind
biases.

Secondly, the development of the dipole in wind differences ahead of the onset of the easterly jet could be related to dif-
ferences in the propagation of equatorial waves, in particular Kelvin waves, between Aeolus and ERA5. Given also that the
wind bias around the the tropical tropopause in Fig. 4b might be attributed to Kelvin wave mixing, it is this hypothesis which
is explored in the next section.

## 3.3 Equatorial waves during the QBO disruption

A number of different methods have been used to identify and isolate equatorial waves from observations, some of which are
discussed by Knippertz et al. (2022). Holton (1973) found that the atmosphere itself acts as a natural bandpass filter which
limits the range of frequencies for which there is a significant Kelvin wave response. Moreover, oceanic Kelvin wave studies
have often used bandpass filtering as a strategy to extract their signal from the large-scale background flow (Polo et al., 2008;
Roundy and Kiladis, 2006), and the decomposition of atmospheric Kelvin waves into different wave period ranges using
bandpass filtering is not without precedent either (e.g. Blaauw and Žagar, 2018; Sjoberg et al., 2017). Figure 6 therefore shows
time-filtered Hövmoller diagrams (where time is placed on the y axis) of the eddy zonal wind (i.e. $u' = u - \overline{u}$) for both Aeolus
and ERA5 at an altitude of 16 km, with the difference between the two in the centre panel. Figure S1 in the supplementary
material shows the raw, unfiltered winds for completeness. A Gaussian band-pass filter is used here to highlight only symmetric
wind structures with a period of 5 to 25 days. As a result, Kelvin wave activity can be seen with varying intensity for the duration
of the disruption, in both Aeolus and ERA5. The dominant waves observed in Fig. 6 have periods in the range 20 to 25 days,
which is at the slower end of the spectrum of all Kelvin waves observed in the atmosphere.

Kelvin waves are known to exhibit a variety of frequency regimes. Convectively-coupled Kelvin waves, such as those anal-
ysed in Outgoing Longwave Radiation (OLR) data from NOAA polar orbiting satellites by Wheeler and Kiladis (1999), tend
to have a faster, narrower range of periods (7 to 10 days). In the TTL and stratosphere, free-travelling waves also occur with
a much broader range of periods (Ern et al., 2008). In general, observational studies split these into three discrete frequency

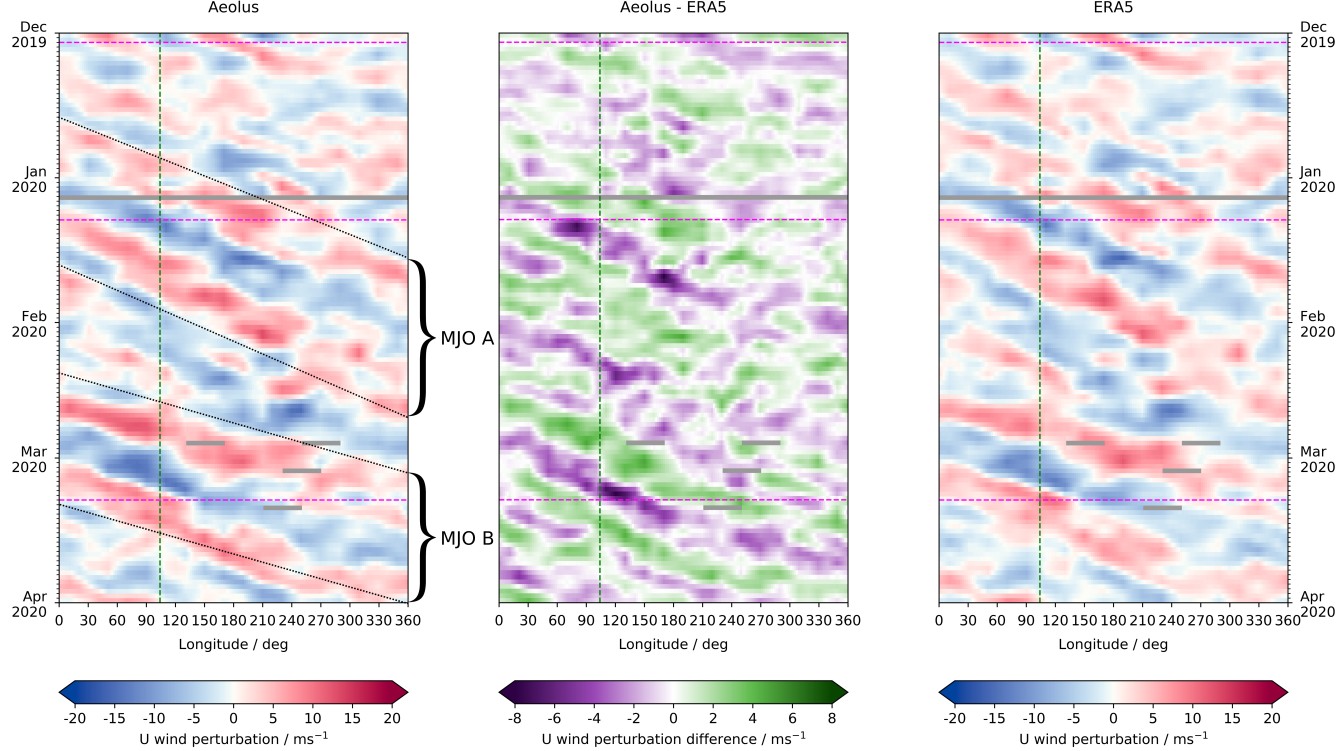

**Figure 6.** Time-filtered Hövmoller plots of zonal wind perturbations from the zonal-mean between 2.5°S and 2.5°N at 16 km for the December 2019 - April 2020 disruption time period, for (a) Aeolus, (c) ERA5 as Aeolus and (b) the difference between (a) and (c). The 3 dashed magenta lines during the period of the disruption correspond to the cross-sectional snapshots in Fig. 7. The vertical dashed green lines mark the longitude of the Singapore radiosonde launch site, 104°. Data is interpolated onto a 2° resolution longitude grid to enable the wave progression to be seen more clearly. Data gaps are marked in grey.

regimes of "slow", "fast" and "ultra-fast" Kelvin waves, which dominate in the lower, middle and upper stratosphere respectively, with the dominant wave phase speed increasing with height (Wallace and Kousky, 1968; Canziani et al., 1994; Forbes
et al., 2009). "Slow", or so-called "Wallace-Kousky" waves, have periods of order 10 to 20 days, and are identifiable around 16 km in both observational (Alexander et al., 2008) and modelling (Žagar et al., 2022) studies. Nonetheless, Shimizu and Tsuda (1997), carried out a periodogram analysis of a radiosonde campaign in Indonesia and found a dominant Kelvin wave period of 20 to 25 days in the 15 to 20 km altitude range, which agrees particularly well with the results here.

Another possible explanation for the slower speed of these waves are the "superclusters" noted by Nakazawa (1988) and
Dunkerton and Crum (1995) amongst others; indeed, the former study identifies several smaller clusters of convective activity which propagate embedded within a larger tropical intraseasonal oscillation (TIO), while the latter alludes to anomalies with periods of up to 15 days traversing within a TIO with a period of 30 to 60 days. Such a long-period oscillation is today termed the Madden-Julian Oscillation (MJO), and it is likely that the two larger-scale pulses in amplitude seen in Fig. 6 form the

active phases of the MJO (marked 'MJO A' and 'MJO B'), within which these eastward propagating Kelvin waves can be
seen. Kikuchi et al. (2018) and Roundy (2020) examine the relationship between the MJO and convectively-coupled equatorial
waves (CCEW) and propose in a similar way that Kelvin waves can act almost as building blocks for the larger MJO envelope,
a picture that matches the phase progression of the waves in this figure.

In Fig. 6b the difference between Aeolus and ERA5 can be seen, showing relatively small differences maximising around 5
to 10 m s$^{-1}$, which is in line with expectations from the random error between the two datasets observed earlier. Aeolus winds
are generally stronger than ERA5 winds in each zonal wind pulse, consistent with a pattern that exhibits the same Kelvin wave
behaviour; and typically the timing of each phase in Aeolus slightly precedes the same in ERA5. Overall, this analysis shows
that whilst perhaps weaker than in the case of the 2015/16 disruption, as alluded to by Kang and Chun (2021), much Kelvin
wave activity during the mature phase of the 2019/2020 disruption is still easily observable, here by Aeolus.

Figure 7 shows three vertical along-equator cross-sections of the eddy zonal wind for Aeolus. For ease of comparison with
the wind profiles at Singapore in Fig. 3, the green dashed lines demarcate the longitude of the Singapore radiosonde launches
and co-located Aeolus and ERA5 profiles. Each cross-section corresponds to the dashed magenta lines seen in both Fig. 1 and
Fig. 4. These dates are chosen firstly to represent an even temporal spread across the QBO disruption, secondly to match the
snapshots in Fig. 3 as closely as possible, and thirdly yet most importantly, since the vertical structure of an equatorial Kelvin
wave can be seen in each of them.

For all three dates, there is a strong dipole in the raw zonal wind centered around 180° longitude (not shown), likely
corresponding to divergence associated with the upwelling maximum of the Walker circulation in the west Pacific ocean. This
feature can be seen in Fig. S1 in the supplementary material, with a dipole centred around the same longitude. Upper-level
horizontal divergence is at a maximum here, and remains as a quasi-stationary feature in similar cross-sections during this
period (also not shown). To the west of this, in Fig. 7a, b and c, the filtered winds show the diagonally slanted pattern of
a Kelvin wave, maximising in amplitude around the longitude of the maritime continent, between ∼(95 to 155)°E. The fine
vertical resolution offered by Aeolus shows a strong vertical gradient in the zonal wind around the tropopause where the Kelvin
wave amplitude is greatest, particularly in Fig. 7b and c. The wave-like structure of the filtered wind profile between 15 and 22
km at the longitude of Singapore in the first cross-section (Fig. 7a) matches the corresponding raw wind profile a day later in
Fig. 3d quite well. The general morphology of this snapshot is similar to the following two cross-sections (Fig. 7b and c), but
the Kelvin wave amplitude at these later dates is stronger, matching the same behaviour seen in Fig. 6.

The data here is interpolated onto a finer grid in longitude which enables the wave to be seen more clearly, however the
coarseness of the original data is still visible. This coarseness is caused by the horizontal spacing between each Aeolus overpass
in the tropics, such that with 16 complete orbits there are 32 individual overpasses each day (ascending and descending nodes),
separated by an average of around 12° longitude. As a consequence, Aeolus' analysis of equatorial waves in the tropics is
limited by this constraint, which could prove a challenge if attempting to investigate smaller-scale features involved in the
dynamics of the QBO disruption.

In order to better understand how Kelvin wave activity is modelled differently in ERA5 compared with observations by
Aeolus, it is useful to look at the average wind perturbations over the disruption time-frame. Figure 8a shows a composite of

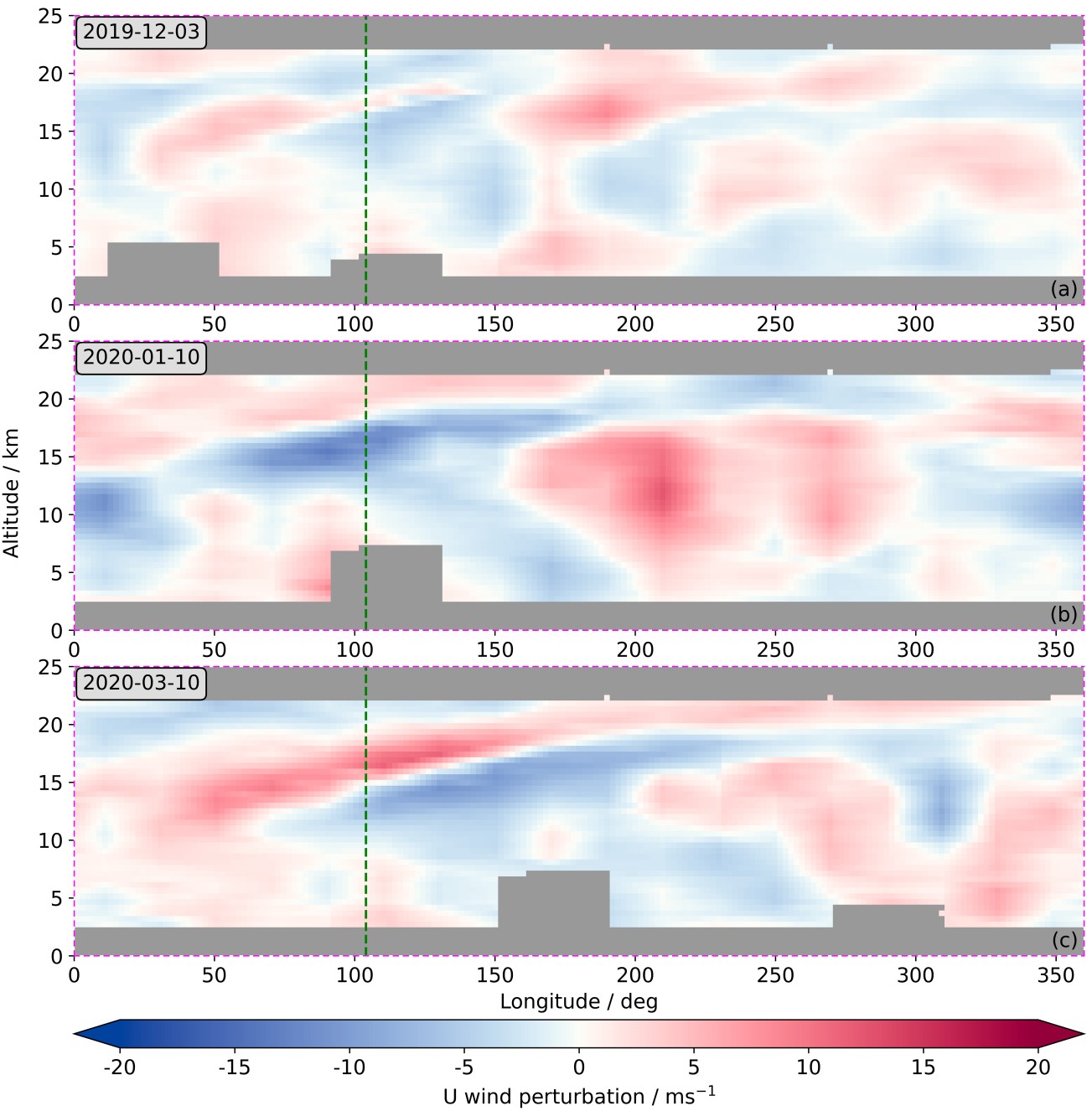

**Figure 7.** Vertical along-equator cross-sections of the daily averaged Kelvin wave-filtered zonal wind perturbations after removing the zonal-mean, between 2.5°N and 2.5°S, for (a) 2019-12-03, (b) 2020-01-10, and (c) 2020-03-10. Each figure corresponds to a vertical slice marked by a dashed magenta line in Fig. 6. Data is interpolated onto a 2° resolution longitude grid to enable the Kelvin wave structure to be seen more clearly. The vertical dashed green line marks the longitude of the Singapore radiosonde launch site, 104°E, as in Fig. 6.

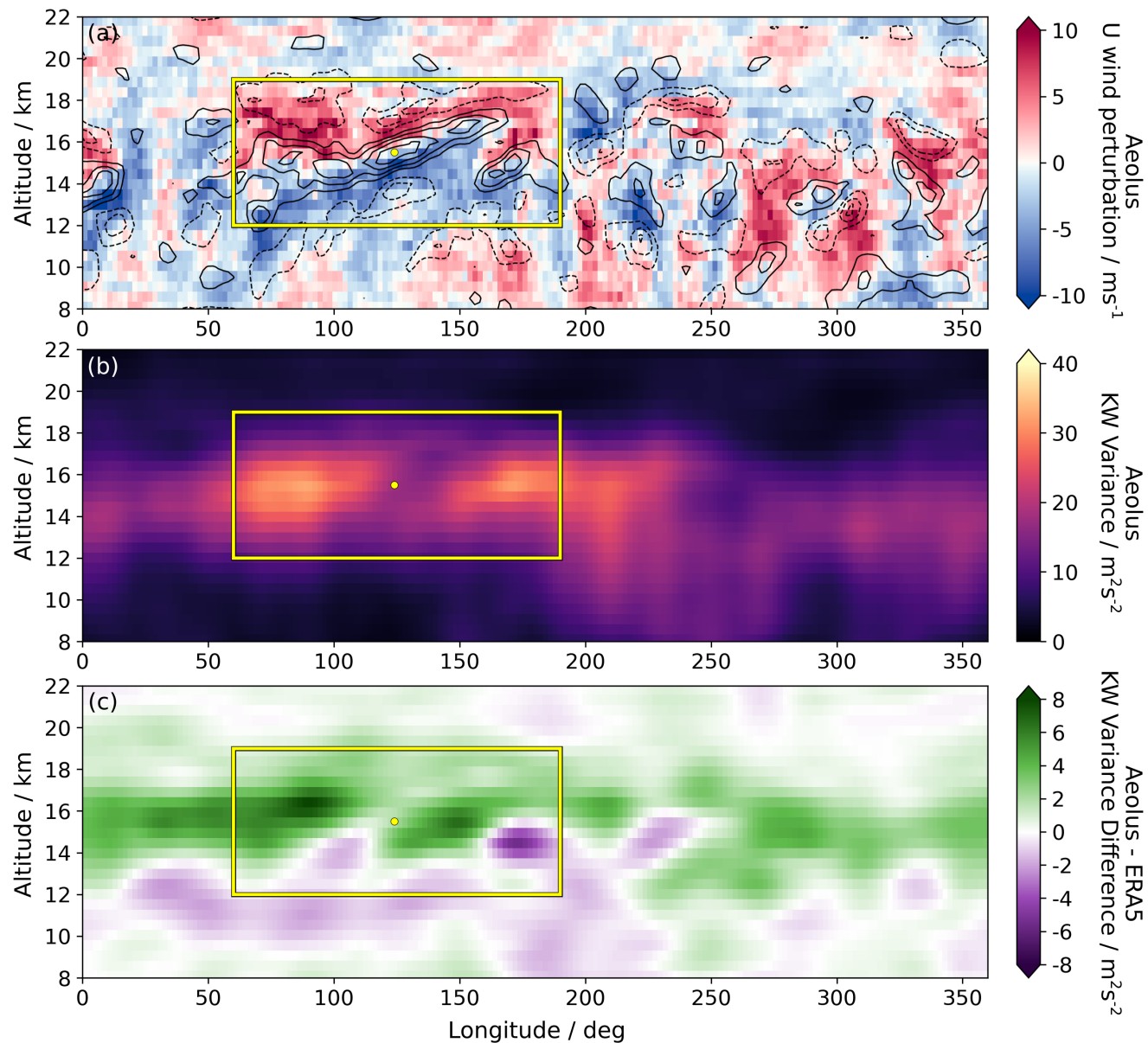

**Figure 8.** Vertical structure of (a) the median Kelvin wave-filtered zonal wind perturbations from Aeolus for the December 2019 - April 2020 disruption time period (coloured) and the corresponding vertical wind shear (black contours at 0.002 s$^{-1}$ intervals), (b) the temporal variance of the above perturbations and (c) the difference between the variance for Aeolus and ERA5, between 2.5°S and 2.5°N. The yellow dot marks the location of the maximum wind shear in panel (a): 124°E at 15.5 km. The yellow box surrounds the key area of interest where the highest Kelvin wave amplitudes are, between 12 and 19 km, and between 60°E and 190°E.

the vertical structure of the Kelvin wave-filtered winds between December 2019 and April 2020, along with the corresponding vertical wind shear. This calculation was performed by taking the median of the daily wind perturbations at each longitude and altitude and then normalising by the median RMS of the entire domain for each day. The wind shear is determined by taking the difference between each level and then scaling in a similar fashion. The yellow box highlights the region of strongest vertical wind shear, maximising at $(+0.0097 \pm 0.0028)$ s$^{-1}$ at an altitude of 15.5 km and a longitude of 124°E, which is marked by the yellow dot. This contrasts with ERA5 which has a wind shear maximum of only $(+0.0045 \pm 0.0026)$ s$^{-1}$ at an altitude of 14.5 km and a longitude of 112°E (not shown). This value is ~50% lower than is found for Aeolus. These results agree well with Houchi et al. (2010) which found peak median vertical wind shear values of 0.008 s$^{-1}$ in radiosonde data and 0.005 s$^{-1}$ in the ECMWF operational model analysis fields in the tropics. While studying thin cirrus in the TTL, Jensen et al. (2011) represented moderate shear with a value of 0.005 s$^{-1}$ and relatively strong TTL wind shear with 0.015 s$^{-1}$, suggesting that the Kelvin waves present during the evolution of the QBO disruption produced moderate vertical wind shear conditions quite frequently.

Crucially, Fig. 8b shows that the region of highest vertical wind shear corresponds to the area where Aeolus observes the greatest Kelvin wave variance. This maximises around 15 km in altitude; somewhat lower than observed by Scherllin-Pirscher et al. (2017) using GPS-RO measurements, but consistent with Ern et al. (2023) using Aeolus. Kelvin waves preferentially occur in the eastern hemisphere, in particular over the Indian and West Pacific oceans; the same as was found by Bergman and Salby (1994) in synoptic Global Cloud Imagery (GCI). Due to the upward branch of the Walker circulation, winds are predominantly easterly in the eastern hemisphere and westerly in the western hemisphere. This dipole pattern has been found by Kawatani et al. (2009, 2010a, b) and Flannaghan and Fueglistaler (2013) to inhibit the propagation of a large part of the Kelvin wave spectrum in the western hemisphere due to critical-level filtering by the westerlies aloft. The latter of those studies suggested that Kelvin wave variance in the troposphere is driven mostly by the the zonal winds in the TTL, rather than the climatology of tropospheric wave sources. There is also likely to be a measure of Doppler shifting by the ambient flow, as noted by Yang et al. (2012). Also seen in Fig. 8b is a slight decrease in Kelvin wave variance between 110 and 140°E over the maritime continent, a result observed by Flannaghan and Fueglistaler (2012) as well. Furthermore, the peak over the western Pacific has been suggested by Ryu et al. (2008) to be particularly relevant for the issue of stratosphere-troposphere exchange, whereby the fluxes of tropospheric air entering the stratosphere are enhanced due to a relatively small, confined, cold region associated with the upper-level divergence at the centre of the Walker circulation dipole. This could be an important process to consider in the context of the apparent reanalysis bias shown in Fig. 4b.

The peak magnitudes of the Kelvin wave variance range from 30 to 40 m$^2$s$^{-2}$, consistent with the findings in Ern et al. (2023). That study also used Aeolus data; however, their focus was primarily on analysing the temporal variability of equatorial waves during the Aeolus mission timeframe. In order to assess potential disparities between the vertical and longitudinal structure of Kelvin waves observed by Aeolus and those represented in ERA5, Fig. 8c is plotted. This reveals a pronounced band of increased variance in Aeolus data compared with ERA5, at the altitude associated with the highest Kelvin wave variance. At 15 km, Aeolus' Kelvin wave variance is generally 3 to 6 m$^2$ s$^{-2}$ higher than ERA5. A similar vertical structure is also observed in the zonal-mean, as documented over an extended time span in Ern et al. (2023). Notably, in the region where the differences

are the greatest, also within the same yellow box, there is a double dipole structure corresponding to the two variance peaks in Kelvin waves over the Indian and West Pacific Ocean. This indicates differences in both the location and magnitude of these variance peaks when comparing Aeolus to ERA5, with the maxima in Aeolus occurring above and to the west of those in ERA5. Žagar et al. (2021) found that in the ECMWF operational model, run with and without assimilated Aeolus data, the greatest differences in the representation of Kelvin waves occurred where wind shear was highest; which agrees well with the results here.

Given the observed differences in the altitude of the peak in Kelvin wave variance, as well as the apparent longitude offset in the two peaks in Fig. 8b and c; and combined with the differences observed between Aeolus and ERA5 in the Hovmoller plot in Fig. 6b, it is possible that both the speed and vertical depth of the waves is being captured differently in ERA5 to Aeolus observations of the real atmosphere. If there are certain parts of the Kelvin wave spectrum that are not being well represented by the reanalysis model, that could explain the discrepancy found here. This question is investigated further in the next section by considering power spectra of these waves, in the same form as was used by Wheeler and Kiladis (1999).

### 3.4 Power spectra of tropical winds

To produce these power spectra, the method of Salby (1982) was employed, in the same way as was done by Alexander et al. (2010) for equatorial wave analysis using the High Resolution Dynamics Limb Sounder (HIRDLS) instrument. Salby (1982) describes a method to analyse equatorial waves in asynoptically sampled data from a polar orbiting satellite, of which Aeolus is an example. To deal with temporal inconsistencies caused by the orbit of the satellite, the ascending and descending nodes are treated separately. In the case of this study, Aeolus HLOS winds are binned by altitude into 3 km-deep overlapping bins, by latitude into bins 2.5° wide, and by orbit number; before the wind perturbations are then found by subtracting the daily zonal-mean wind. Data gaps are filled by linear interpolation using the surrounding orbits at the same latitude and altitude, with consecutive gaps being filled progressively from the centre of the gap outwards. 1D fast Fourier transforms (FFTs) are then performed on each node independently, before the output is combined to form wave spectra as a function of zonal wave number $k$ and frequency $\omega$, as in Salby (1982). In the same way as in Alexander et al. (2010), this method does not resolve frequencies $\omega > 1$ day$^{-1}$, or wavenumbers $|k| > 8$. Due to the higher noise in Aeolus data than, for example, HIRDLS, the FFTs must be calculated over a sufficiently long time period to extract the signal from the noise. Here, to study the waves during the disruption in particular, data from 13 November 2019 to 30 June 2020 is used. This time period is also constrained partly by data quality and availability.

To begin with, the symmetric component is found by calculating the mean spectrum for each pair of latitude bins. This is then removed from the spectrum for one of the hemispheres to find the antisymmetric component for that particular latitude. Figure 9 shows, on the top row, the symmetric component at the equator and the altitude bin centred at 18 km; and on the bottom row, the antisymmetric component at 7.5°S and 19 km. The dispersion curves for Kelvin, mixed Rossby-gravity and n=1 equatorial Rossby waves have been plotted on the spectra with equivalent depths, $h_e$ of 12, 25 and 50 m. In order to compare Aeolus with ERA5, the spectra for each is plotted on the left and right respectively, with the scaled difference between them in the centre.

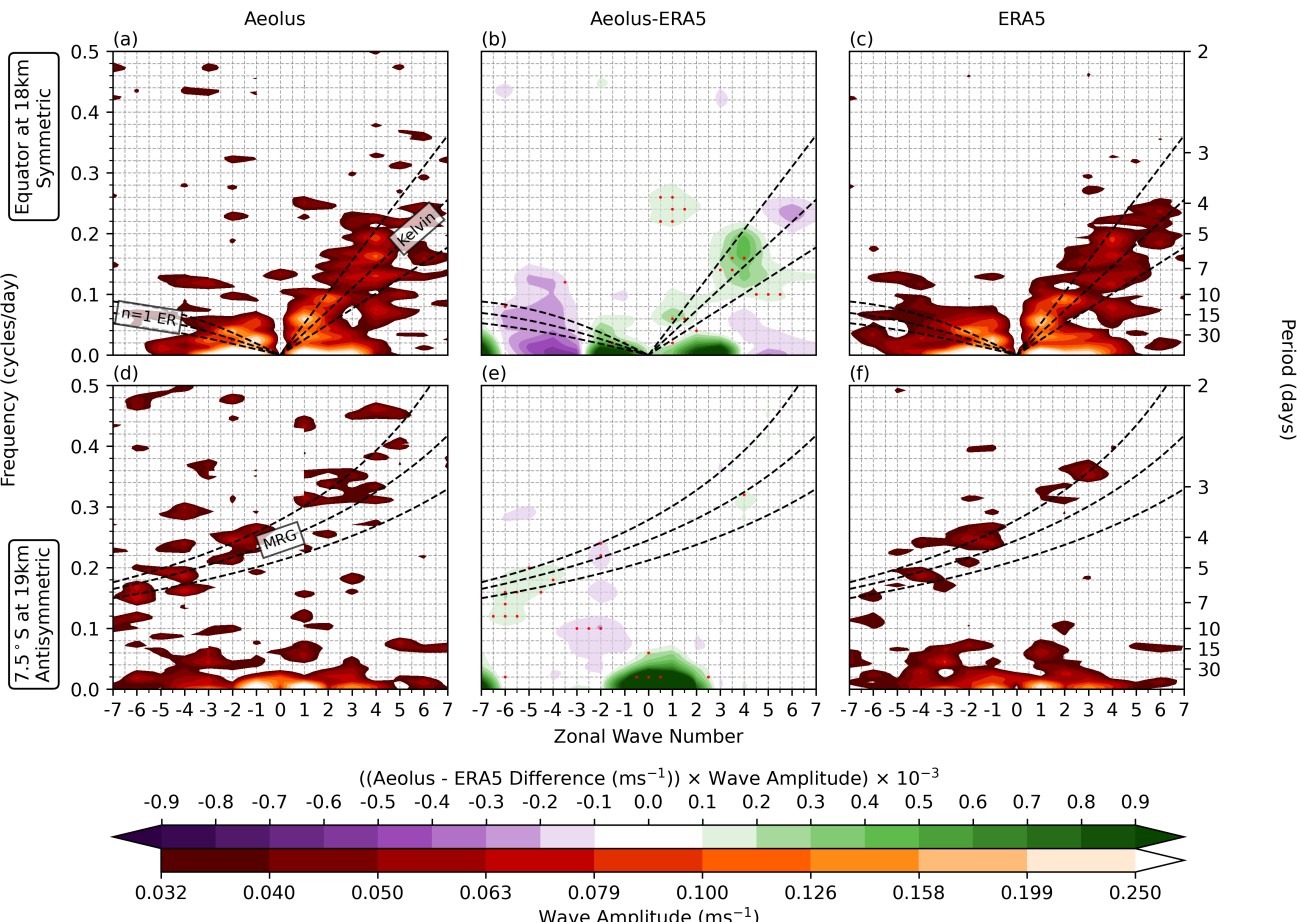

**Figure 9.** Wavenumber-frequency power spectra for the (top) symmetric component, at 18 km altitude at the equator and (bottom) antisymmetric component, at 19 km at 7.5°S, for (left) Aeolus, (right) ERA5 and (middle) Aeolus - ERA5. The middle panels are scaled by the wave amplitude of the Aeolus spectra for each $k, \omega$. Also plotted are theoretical dispersion curves for Kelvin waves, n = 1 equatorial Rossby (ER) waves and mixed Rossby-gravity (MRG) waves with equivalent depths $h_e = 12$ m, 25 m and 50 m. Stippling in the middle panels denote regions where the raw difference is greater than the standard deviation of its spectra i.e. SNR > 1, where the scaled difference after removing noise is greater than $0.1 \times 10^{-3}$, and where p < 0.001 across a $(\pm k, \pm 0.05\omega)$ sampling window.

The dominant feature of the raw difference between Aeolus and ERA5 is that Aeolus exhibits a higher amplitude across the majority of the spectrum for both the symmetric and antisymmetric components (not shown). This is partly due to the higher noise in the Aeolus spectra. Therefore, to aid a direct comparison, this noise has been removed by deducting the median of the original spectral differences in Fig. 9b and e from both these (so that they are centred around zero) and the spectra in panels a and d respectively. These removed noise values are 0.013 m s$^{-1}$ for Fig. 9a and 0.016 m s$^{-1}$ for Fig. 9d. The logarithmic colour scale for Fig. 9a, c, d and f has been optimally adjusted to best show the signature of the waves in both Aeolus and ERA5 through the background noise, and by only plotting amplitudes > 0.032 m s$^{-1}$. Peaks in spectral amplitude can be observed in the regions corresponding to Kelvin and equatorial Rossby waves in Fig. 9a and c, and to mixed Rossby-gravity waves in Fig. 9d and f. This shows that both Aeolus and ERA5 capture these three types of waves during the QBO disruption time-frame, although the spectral power of the mixed Rossby-gravity waves is relatively weak compared with the background in both datasets.

In order to focus on the parts of the spectrum with the highest SNR, and therefore emphasise the spectral components associated with distinct equatorial wave types, the differences are scaled by the mean amplitude of the two datasets at each $k$ and $\omega$. Significance at the p < 0.001 level is denoted by stippling on the figures. In Fig. 9b Aeolus generally exhibits a positive difference in Kelvin wave amplitude compared with ERA5, with some statistically significant regions. At higher $k$ and $\omega$, ($k > 5$, $\omega \sim 4$ days), ERA5 has the greater amplitude, although stippling is absent, indicating lower significance. Elsewhere, there is a dipole in the differences for equatorial Rossby waves with Aeolus demonstrating higher amplitudes at lower $k$ while ERA5 has a higher amplitude at higher $k$; although the lack of stippling suggests low significance for each. There is also a small positive difference in the Aeolus spectra in the centre of Fig. 9b, however this is not associated with a typical wave mode. In the antisymmetric spectra, Aeolus continues to exhibit higher amplitudes at low $k$ and $\omega$, showing some significance. In the region where mixed Rossby-gravity waves are observed, the signal is variable and weak, suggesting little difference in the representation of mixed Rossby-gravity waves between Aeolus and ERA5.

Given the observed differences in Fig. 8c varied with altitude, and since the analysis of Fig. 9 is constrained to two select cases, it is useful to see how the wave spectra vary with height in the TTL across the entire equatorial region. Figure 10 shows this for 16 to 19 km, between 15°S and 15°N. The primary difference between Aeolus and ERA5 is again the higher background noise in the raw Aeolus spectra (not shown), and so this has been removed in the same way as for Fig. 9. For Fig. 10a, d, g and j, the removed noise values are 0.016 m s$^{-1}$, 0.014 m s$^{-1}$, 0.017 m s$^{-1}$ and 0.015 m s$^{-1}$ respectively. After this, there are still some significant positive differences in the range $-2 \leq k \leq 2$ in the middle panels (Fig. 10b, e, h and k), especially at 17 km. The differences at higher frequencies in this wavenumber range are likely to be caused by small experimental biases, and so this feature is not considered in our equatorial wave analysis.

The higher amplitudes in Aeolus at lower $k$ for both Kelvin and equatorial Rossby waves are similar to those seen in Fig. 9b, although there is variation between different altitudes. The region of negative differences for equatorial Rossby waves in the range $-6 \leq k \leq -3$ in Fig. 9b is much weaker in Fig. 10, and not significant. In constrast, more pronounced negative differences emerge for faster, higher $k$ Kelvin waves, although the lack of stippling suggests their limited significance at the p < 0.001 confidence level. This region does however show subtle variations with altitude. At 16 and 17 km Aeolus has lower

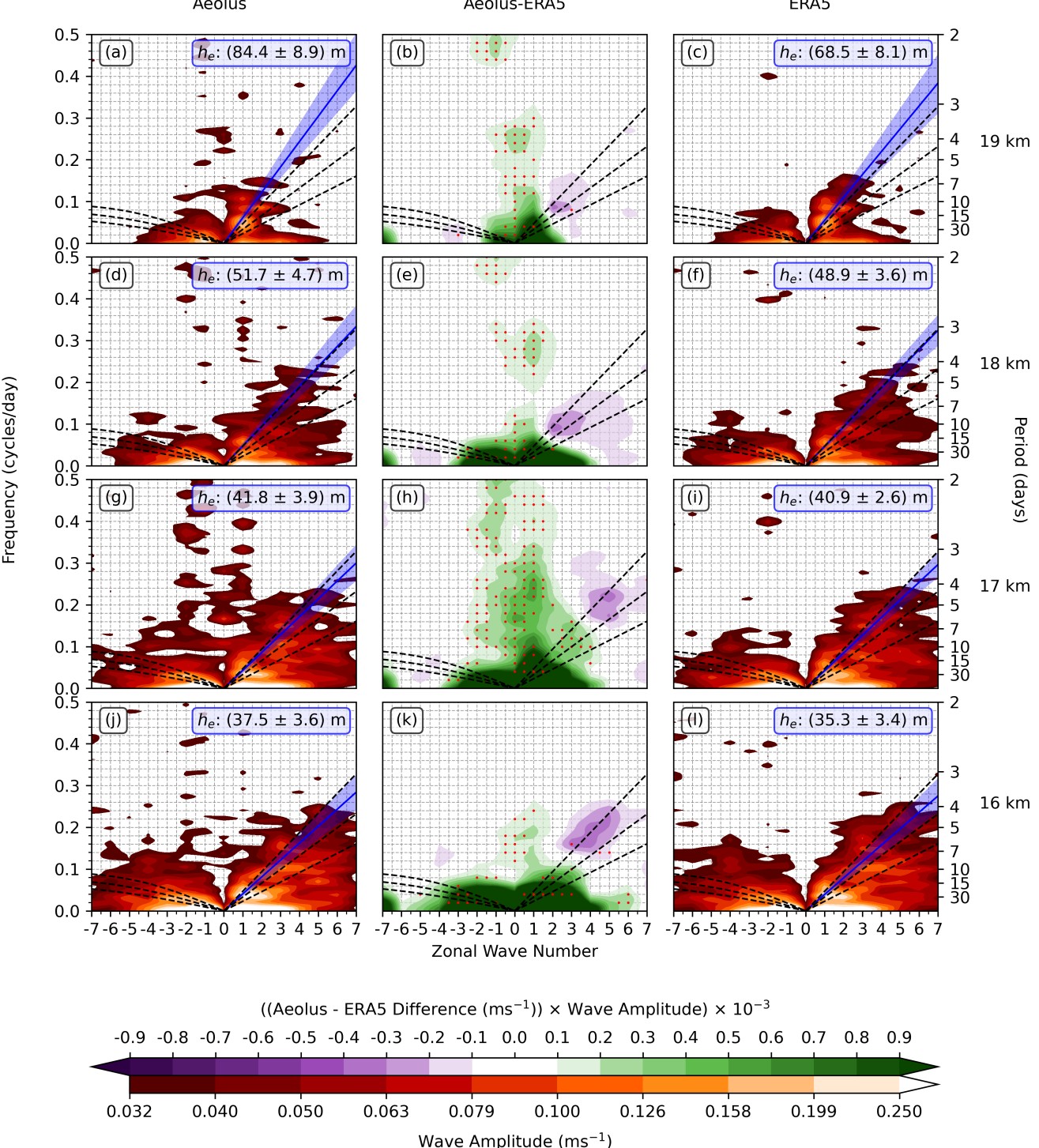

**Figure 10.** Wavenumber-frequency power spectra for the symmetric component between 15° N and 15° S at altitudes of (a, b, c) 19 km, (d, e, f) 18 km, (g, h, i) 17 km and (j, k, l) 16 km, for (left) Aeolus, (right) ERA5 and (middle) Aeolus - ERA5. Theoretical dispersion curves and stippling are plotted the same as in Fig. 9. The best-fit equivalent depth $h_e$ of the Kelvin wave spectra peaks is plotted in blue using linear regression, and is bounded by the shaded 99.9% confidence interval, derived using bootstrapping with 100,000 samples.

amplitude Kelvin waves than ERA5 with periods of 4 to 7 days, whereas at 18 and 19 km, this region of negative differences has drifted to periods of 7 to 15 days. Given the vertical structure observed in Fig. 8c, it is likely that this pattern corresponds to stronger convectively-coupled Kelvin wave variance in ERA5 below 18 km, with the slower free-travelling Kelvin waves remaining higher in Aeolus observations.

The equivalent depth of an equatorial wave is a theoretical concept used to characterise the speed and scale of equatorial waves in the atmosphere. Matsuno (1966) provided a derivation of the different equatorial wave types using shallow water theory, which yields the following dispersion relation between $\omega$ and $k$, as described by Kiladis et al. (2009):

$$\frac{\sqrt{gh_e}}{\beta}\left(\frac{\omega^2}{gh_e} - k^2 - \frac{k}{\omega}\beta\right) = 2n+1, \qquad n = 0, 1, 2, ... \tag{2}$$

where $g$ is the gravitational acceleration, $\beta$ is the meridional gradient of the Coriolis parameter and $n$ is the order of the solution.

Crucially, the equivalent depth, $h_e$, is related to the vertical wavelength of the equatorial waves by the Brunt–Väisälä frequency $N$ and the eigenvalue $\lambda$ as:

$$h_e = \frac{N^2}{g\lambda}, \tag{3}$$

such that, as derived by Wu et al. (2000) and Kawatani et al. (2010a):

$$m = \frac{2\pi}{L_z} = \sqrt{\left(\frac{N^2}{gh_e} - \frac{1}{4H^2}\right)}, \tag{4}$$

where $m$ is the vertical wavenumber, $L_z$ is the vertical wavelength and $H$ is the scale height.

This relationship between $h_e$ and $L_z$ motivates a deeper analysis of the change in equivalent depth with height, in order to see if there are differences which might affect the development of the QBO disruption. To do this, the equivalent depth corresponding to the dominant Kelvin wave mode for all $k > 0$ and $0 < \omega \leq 0.3$ cycles day$^{-1}$ is found from each power spectra using linear regression. Bootstrapping is used to give an estimate of the uncertainty in the values for $h_e$ by resampling the distribution of peak amplitudes 100,000 times. These values are labelled on each spectra in Fig. 10, and are listed with the corresponding values of $L_z$ in Table 1. Values of the Brunt–Väisälä frequency and the scale height are estimated at $N \approx 0.02$ rad s$^{-1}$ and $H \approx 7$ km respectively.

As expected from theory, the dominant equivalent depth of the Kelvin waves is observed by both Aeolus and ERA5 to increase with altitude. For example, Aeolus sees a doubling in $h_e$ between 17 and 19 km from 41.8 to 84.4 m. Garcia and Salby (1987) demonstrated that by having complex vertical wavenumbers, vertically propagating Kelvin waves decay as they move away from their source, an attenuation which varies inversely with group velocity. This altitude dependence arises from the fact that Kelvin waves with shorter vertical wavelengths exhibit slower phase speeds and are influenced more by dissipation processes and critical wind levels (Coy and Hitchman, 1984; Maury and Lott, 2014). Das and Pan (2016) used GPS-RO to

| Altitude / $z$ (km) | Equivalent Depth / $h_e$ (m) | | Vertical Wavelength / $L_z$ (km) | |
|---|---|---|---|---|
| | Aeolus | ERA5 | Aeolus | ERA5 |
| 16 | $37.5 \pm 3.6$ | $35.3 \pm 3.4$ | $6.0 \pm 0.3$ | $5.9 \pm 0.3$ |
| 17 | $41.8 \pm 3.9$ | $40.9 \pm 2.6$ | $6.4 \pm 0.3$ | $6.3 \pm 0.2$ |
| 18 | $51.7 \pm 4.7$ | $48.9 \pm 3.6$ | $7.1 \pm 0.3$ | $6.9 \pm 0.3$ |
| 19 | $84.4 \pm 8.9$ | $68.5 \pm 8.1$ | $9.1 \pm 0.5$ | $8.2 \pm 0.5$ |

**Table 1.** Values of $h_e$ and $L_z$ at different altitudes for Aeolus and ERA5.

investigate the changes in Kelvin wave activity during El Niño events, and found vertical wavelengths of $\sim 6$ km for 24-day Kelvin waves and $\sim 8$ km for 16-day Kelvin waves in the lower stratosphere, broadly agreeing with the results in Table 1. Gahtan and Tian (2022) found a similar result using Atmospheric Infrared Sounder (AIRS) measurements. They suggested that this shift towards higher frequencies and larger vertical wavelengths, accompanied by increased equivalent depths, as Kelvin waves propagate upwards, is influenced by the impact of differential damping from Newtonian cooling.

Comparing Aeolus with ERA5 in Fig. 10, the observed differences in the dominant equivalent depth are mostly within 1 standard deviation. However, the Kelvin waves observed by Aeolus exhibit consistently higher equivalent depths, and this is a difference that widens with altitude. On average, Kelvin waves in ERA5 have equivalent depths $\sim 8\%$ lower, and vertical wavelengths $\sim 4\%$ lower than Aeolus. Although this difference is relatively low, it is consistent with the findings so far in this study that suggest the deficit in vertical wind information assimilated into atmospheric reanalyses causes biases in both wind shear and the vertical wavelength of Kelvin waves; compared with the real atmosphere as observed by Aeolus. The lower equivalent depth of the Kelvin waves in ERA5 suggests that the reanalysis retains waves with smaller vertical wavelengths to higher altitudes, whereas in Aeolus observations, these waves tend to either dissipate due to Newtonian cooling or break at critical wind levels. The implications of this disparity between ERA5 and Aeolus on the development of the QBO disruption are hereby discussed in Sect. 4.

## 4 Discussion

The QBO disruption of 2019/2020 consisted of a weakening of the westerly winds in the lower stratosphere, and the development of an easterly jet around an altitude of 22 km. This study has explored a disparity in the timing of the onset of this easterly jet between Aeolus observations and ERA5, and found differences in TTL winds and Kelvin wave variance which might help to explain this. Such identification of regions where DWL satellites like Aeolus differ from atmospheric reanalyses and operational NWP model forecasts is crucial to improving our understanding of phenomena like the QBO disruption, and could help lead to improvements in future reanalyses and models. Martin et al. (2023) demonstrated a time-varying impact of the QBO 2020 RBS on modelling of the QBO just after the disruption, with beneficial effects in the equatorial stratosphere early in the forecast window. Likewise, Rennie et al. (2021) and Žagar et al. (2021) have already showed substantial improvements to ECMWF forecasts in the TTL region as a consequence of the assimilation of Aeolus data.

The results of this particular study suggest that one contributing factor to these improvements is Aeolus' ability to accurately capture regions of sharp vertical wind shear. A result that is in agreement with Bley et al. (2022). Consequently, Aeolus gives a more realistic representation of Kelvin waves with shorter vertical wavelengths compared to reanalysis and NWP models. Much of the spectrum of Kelvin waves that are generated by convection is reproduced in the reanalysis, or is even slightly enhanced; likely due to the impact of the parameterisation schemes. However, the smaller-scale Kelvin waves, with shorter vertical wavelengths, encounter critical levels, break and dissipate lower down in the TTL in Aeolus observations. This corresponds to the region of higher Kelvin wave variance shown in Fig. 8b and c. As a result, the equivalent depth corresponding to the dominant Kelvin waves is increased relative to the reanalysis. Pahlavan et al. (2021) showed that, in ERA5, westerly accelerations due to resolved waves in the lower stratosphere can be mainly attributed to Kelvin waves. Figure 5 demonstrates the effect of this enhanced Kelvin wave breaking in Aeolus occurring progressively from 16 to 19 km in the TTL, which is a strengthening of the westerly winds between 18 and 21 km. Above this region, at 22 km, ERA5 exhibits a greater deposition of westerly momentum compared with Aeolus. This is because Kelvin waves with shorter vertical wavelengths are propagating to a higher altitude in the reanalysis.

Given that easterly winds in the lower stratosphere have been shown to enhance Kelvin wave propagation in the TTL (Das and Pan, 2016), and even modify the tropopause structure as a consequence (Ratnam et al., 2006; Kedzierski et al., 2016), the westerly bias of the reanalysis at 22 km likely contributes to the weaker Kelvin wave amplitudes observed below. As the disruption easterly jet strengthens, these differences compound over time, until the gradient in the HLOS wind differences eventually reaches 5 to 8 m s$^{-1}$ during early 2020 (Fig. 5). Although this increasing discrepancy is partly caused by inter-seasonal variability, it is likely that the increase in Kelvin wave variance, observed during the later part of the time period shown in the Hövmoller plots in Fig. 6, is related to these stronger easterly winds associated with the QBO disruption. The lag in the change of zonal-mean winds in ERA5 may also contribute to the slight phase lag in Kelvin waves observed here.

Given the wind biases observed near the tropopause in Fig. 4a, b and c, it is likely that the reanalysis is underestimating wind speeds in the UTLS as well as regions of strong vertical wind shear. These results show that a deficit in Kelvin wave breaking at shorter wavelengths may be a contributing factor. Such a reduction in the zonal force provided by eastward propagating waves in reanalysis has been found before, particularly for smaller-scale gravity waves as in Holt et al. (2016). However, it is also possible that Aeolus is overestimating wind speeds in the same region, despite the low systematic biases demonstrated by Abdalla et al. (2020), and that it is therefore a combination of these two factors which leads to the differences observed in this study. Nonetheless, studies concerning the QBO, which use atmospheric reanalyses or NWP models to diagnose the role of equatorial waves, may be affected by this bias in Kelvin wave variance at certain altitudes. Given that operational NWP models poorly predicted the onset of the disruption at the time, it is likely that this under-representation of vertical wind shear, resulting in the reduced breaking of shorter wavelength Kelvin waves, significantly influenced both the real-time forecasts as well as the reanalysis studied here. In addition, issues such as the 'spring predictability barrier' which hinders El Niño forecasts, due in part to measurements of WWBs, some of which can be seen in Fig. 1, may be combatted if their vertical structure is captured more accurately by instruments like Aeolus.

The dominant source of Aeolus error, the random error, should also be considered; which with respect to both the Singapore radiosonde and ERA5 reanalysis, is 9.30 m s$^{-1}$ and 7.58 m s$^{-1}$ respectively for the profiles in Fig. 3. This is especially important in the later analyses, where the eddy component of the zonal wind has been isolated from the zonal-mean zonal wind, and interpreted as a function of longitude. Given the average separation of Aeolus orbits in the tropics is around 12°, the longitudinal resolution of Aeolus data limits this type of analysis somewhat. In spite of this, through either the careful
averaging of data to suit a study's specific requirements, or the assimilation of data into a reanalysis or NWP model, this issue can be overcome. Here, the data has been carefully binned to minimise the impact of these limitations.

Overall, the primary limitation on observing the QBO disruption using Aeolus is the limited vertical extent of the measurements, due to the height range covered by the onboard range-bin settings during this period. Since Aeolus is a demonstration Earth Explorer mission, these settings change regularly in response to calibration/validation analysis, evolving NWP require-
ments and multiple scientific campaigns aimed at the development of a future operational wind lidar platform. Aeolus' designed maximum measurement altitude is 30 km, which was briefly reached during a short period in early 2022 whilst observing the impacts of the Hunga Tonga volcano eruption. Given that much of the QBO's progression occurs in the 20 to 30 km range, there is a need for future operational space-borne DWL instruments to measure winds up to at least 30 km or higher, both for QBO prediction and research. This requirement has been discussed by a number of studies including Wright et al. (2021) and
Banyard et al. (2021). Nonetheless, the weekly QBO 2020 RBS that were introduced onboard Aeolus in June 2019 contributed significantly to observations of the lower portion of the QBO after this date.

Finally, given the long oscillation period of the QBO, and since much of the Aeolus measurement time-frame is dominated by the disruption of 2019/2020, it is difficult to use Aeolus data to study the normal progression of this phenomena, and in particular to answer questions relating to the occurrence of QBO disruptions in the future. Some global climate models infer
an increased frequency of disruptions as a consequence of climate change (Anstey et al., 2021), however with just two such disruptions so far there is a limited sample size of events to study. It may be that disruptions either are or have become part of the natural cycle of the QBO on longer timescales. Kang et al. (2022) suggests that stronger westerly winds in the equatorial lower stratosphere as a consequence of climate change will provide more favourable conditions for QBO disruptions in the future. Although the limited time duration of the Aeolus mission prevents this question from being answered here, future
operational DWL satellites that measure tropical stratospheric winds will be better able to observe such changes.

## 5    Conclusions

In conclusion, this study has investigated the 2019/2020 QBO disruption using novel data from ESA's Aeolus satellite. Aeolus is very well suited to observing zonal wind phenomena in the tropics because of its measurement geometry. It can observe the lower portion of the QBO, and with changes to the onboard range-bin settings made in 2020, can see much more of the
QBO's phase progression in the lower stratosphere. The evolution of the disruption with its anomalous upward propagating easterly jet and overlying westerly winds can be seen using Aeolus, although this is hindered slightly by the satellite's vertical measurement range at the time. The maximum zonal-mean easterly winds within the part of the disruption jet that can be

observed by Aeolus reaches 20 m s$^{-1}$ during July 2020. Aeolus then captures the following downward propagating westerly QBO phase, followed by the next easterly QBO phase at high altitudes, which is aided by settings to observe wind returns from high-altitude aerosol resulting from the Hunga Tonga eruption in January 2022.

There is good agreement between co-located profiles using Aeolus, Singapore radiosonde and ERA5 data, although there exists a shallow negative wind bias of around -3 m s$^{-1}$ near the tropopause which is partly a result of insufficient vertical resolution as well as deficits in the data assimilation of existing satellite data in the reanalysis. The standard deviation of the differences between the datasets is in broad agreement with existing literature, with an average Aeolus-ERA5 random error of 7.58 m s$^{-1}$, although there is greater spread at higher altitudes due to the decreasing Aeolus SNR here. The onset of the easterly disruption anomaly at 22 km occurs on 18 December 2019 in the Aeolus zonal-mean HLOS wind, preceding the wind reversal in ERA5 by 5 days. This discrepancy has two main causes, which are (i) a tendency for the reanalysis to revert to is model climatology due to insufficient observational constraints; and (ii) differences in Kelvin wave propagation due to reduced dissipation of waves with smaller vertical wavelengths within the TTL in the reanalysis.

During the disruption, Aeolus captures slow eastward-propagating Kelvin waves, embedded within larger clusters of westerly zonal wind pulses which are likely associated with the active phase of the MJO. Aeolus winds are generally stronger than ERA5 in each wave pulse, and the phase precedes that of the reanalysis slightly. The vertical structure of the Kelvin waves has been analysed using along-equator cross-sections of the zonal wind, showing diagonally oriented wave fronts and maximum wave amplitudes over the Indian and West Pacific oceans. In these cross-sections, the dipole in zonal wind associated with the Walker circulation is seen to inhibit Kelvin wave propagation in the western hemisphere, but enhance it in the eastern hemisphere. Additionally, Aeolus exhibits Kelvin wave variance that is generally 3 to 6 m$^2$ s$^{-2}$ higher compared with ERA5, particularly around 15 km altitude. This can be matched to a region of higher vertical wind shear which is captured more sharply by Aeolus. Wind shear values maximise at a longitude of 124°E, at (+0.0097 ± 0.0028) s$^{-1}$, which is around twice the wind shear found in ERA5.

Analysis of the equivalent depths of the dominant Kelvin wave mode has shown small differences between Aeolus and ERA5. The latter has equivalent depths that are ~8% lower, and vertical wavelengths ~4% lower. Nonetheless, the larger values observed by Aeolus suggest that ERA5 does not capture as much breaking and dissipation of Kelvin waves with shorter vertical wavelengths, especially within the TTL. This leads to less westerly momentum being deposited between 18 and 21 km in ERA5, just above the region of greatest Kelvin wave variance as measured by Aeolus. The resultant impact on the QBO disruption is reflected in a weaker wind shear in ERA5 at the location of the easterly jet which develops at 22 km; and this ultimately contributes to the 5 day lag in its onset. This mechanism is also likely to play a role in the poor predictability of the onset of the QBO disruption in forecast models.

Finally, some of the limitations of using Aeolus data to analyse the QBO disruption and the dynamics of the tropical stratosphere more generally have been discussed. The primary constraint is the limited vertical extent of the measurements due to the onboard range-bin settings during the period of analysis. However, the random error of Aeolus winds and reduced longitudinal resolution caused by orbital geometry also play a key role in adding uncertainty to the results. In spite of these, Aeolus and future DWL satellites show a lot of promise in observing events like the 2019/2020 QBO disruption, and their contribution to

future reanalyses and operational NWP models is likely to improve our understanding of the mechanisms behind such events even further.

*Author contributions.* TPB - Conceptualization, Data curation, Formal Analysis, Investigation, Methodology, Project administration, Software, Validation, Visualization, Writing – original draft, Writing – review & editing.

CJW - Conceptualization, Data curation, Funding acquisition, Investigation, Methodology, Project administration, Resources, Software, Supervision, Writing – original draft, Writing – review & editing.

SMO - Conceptualization, Methodology, Writing – review & editing.

NPH - Conceptualization, Methodology, Software, Supervision, Validation, Writing – review & editing.

GH - Conceptualization, Writing – review & editing.

LC - Conceptualization, Writing – review & editing.

PAN - Conceptualization, Writing – review & editing.

NB - Conceptualization, Writing – review & editing.

MB - Conceptualization, Software, Methodology.

MJA - Conceptualization, Software, Methodology.

*Competing interests.* The authors declare that they have no competing interests.

*Acknowledgements.* The authors would like to thank the Aeolus DISC team for supporting this study and for their helpful communications throughout. We would like to acknowledge Shwei Lin Wong of the Meteorological Service of Singapore for information about the radiosonde data. We also thank George Kiladis for his useful advice on the power spectra analysis. This work used JASMIN, the UK's collaborative data analysis environment (https://jasmin.ac.uk) (Lawrence et al., 2013). T. P. Banyard is funded by Royal Society grant RGF/EA/180217 and EPSRC grant EP/R513155/1. C. J. Wright is funded by Royal Society grant RF/ERE/210079 and NERC grant NE/V01837X/1. C. J. Wright and N. P. Hindley are funded by NERC grant NE/R001391/1 and NE/S00985X/1.

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
