# Peer review of "Aeolus wind lidar observations of the 2019/2020 Quasi-Biennial Oscillation disruption with comparison to radiosondes and reanalysis"

_EGUsphere, 2023_

## Referee Comment (RC1)

Referee comment on

**Aeolus wind lidar observations of the 2019/2020 Quasi-Biennial Oscillation disruption with comparison to radiosondes and reanalysis**

egusphere-2023-285

This study provides a good investigation of the ability of the Aeolus data to contribute to the understanding of the QBO dynamics. To this end, Aeolus observational data from various range-bin settings were compared with ERA5 reanalyses and high-quality radiosonde observations. The results and discussions highlight both opportunities and limitations of the Aeolus DWL mission. The manuscript is well written, concise, and includes appropriate figures. I recommend publication after the following minor issues and suggestions are resolved or considered.

**2. Data and Methods**

- I think the order of the data description could be changed. In the subsection about the radiosonde data, there is already written about Aeolus measurement geometry, overflights and a special Aeolus setting. It would be easier for the reader to understand this if he/she has learned about Aeolus before.

**2.2. Aeolus**

- Somewhere it should be mentioned that Aeolus is a polar orbiting satellite.
- Line 72: Please consider making two sentences out of this (The satellite's orbit is sun-synchronous with 15.6 orbits each day and a repeat cycle of 7 days. For the duration of the observing period in this study there is a close overpass to the site of the Singapore rawinsonde station between 22:55 and 23:00 UTC every Wednesday.)
- Consider offering the reader a reference for more detailed information about the Aeolus data products and processing (e.g., Reitebuch et al., 2018 and Tan et al. 2008).
- Are there important changes between Baseline 11 and Baseline 14 that the reader should know about?
- What are random error sources that could be affect the data quality during the analyzed period?
- Please add a sentence as to why only Rayleigh winds are considered. In the upper troposphere in the tropics, I would expect good quality Mie winds at cloud top level or within thin clouds.

**2.3. ERA5**

- Consider including the formular how to calculate ERA5 HLOS wind.
- Please provide some information on the quality of the ERA5 data in the upper troposphere/lower stratosphere. NWP models typically have large uncertainties at these altitudes that could potentially affect the results(?).

**3. Results**

**3.1 Aeolus observations of the QBO disruption**

- Line 168: repeating information (see Line 129)

**3.2. Validation against reanalysis and radiosondes**

- Line 208: I'm not totally convinced, that ERA5 stratospheric wind errors are that small (due to large model errors at these levels). E.g., the warm bias is mentioned later in the text, what makes this statement slightly doubtful.
- Line 235/236: Is there any conjecture about this height dependence of the Aeolus bias?

**3.3. Equatorial waves during the QBO disruption**

- Please add a reference about the wave filtering method that is used here.
- Figure 6: What are the thick horizontal gray lines (early January 2020 and around March 2020)? I can't find a description in the caption or the text.

**Introduction/Discussion**

Regarding the current state of Aeolus research consider adding the reference Martin et al., 2023 (https://doi.org/10.5194/wcd-4-249-2023) about the impact of assimilating Aeolus observations on QBO (with and without QBO RBS) and ENSO in 2020 in the Introduction or Discussion

**Technical stuff**

- I believe the EGU guidelines say that there should be a space in between for the unit **m s$^{-1}$** .
- Sometimes you use **2019/2020,** sometimes **2019-2020** or **2019-20**. Please make this consistent.
- Line 83: please change **2,000 m** --> **2.000 m**
- Line 84: please write out the abbreviation UTLS once
- Line 105: please write out the abbreviation ECMWF once
- Line 155: please write out the abbreviation GNSS-RO once
- Line 272: please write out the abbreviation OLR once
- Line 358: abbreviation DWL is already defined in Line 174

---

## Author Comment (AC1)

**Response to Reviewer 1**

**Overview**

We thank the reviewers for their helpful comments on our study. Based on their comments, the manuscript has been substantially improved since its initial submission. We briefly summarise the improvements here in an overview, before responding to the reviewer's specific comments.

The key improvements to the paper include, but are not limited to:

- A new section investigating equatorial waves, in response to comments by reviewers #2 and #3. This includes (a) new analyses of Kelvin wave variance during the QBO disruption (b) new analyses of the symmetric and antisymmetric power spectra, and (c) a consideration of the equivalent depth $h_e$ and vertical wavelength $L_z$ of Kelvin waves. These results help to clarify equatorial wave processes during the disruption, are consistent with our existing material and other studies, and have improved and expanded the scope of the our study.
- Statistical tests which have been applied to our results, including to the comparison of Aeolus with ERA5 and radiosonde measurements, and to our calculation of the equivalent depth.
- A more comprehensive description of important Aeolus and ERA5 biases which might influence the results in this study, with additional information on these for the reader's benefit.

We now respond to the specific comments by reviewer 1 below.

**Reviewer 1**

*This study provides a good investigation of the ability of the Aeolus data to contribute to the understanding of the QBO dynamics. To this end, Aeolus observational data from various range-bin settings were compared with ERA5 reanalyses and high-quality radiosonde observations. The results and discussions highlight both opportunities and limitations of the Aeolus DWL mission. The manuscript is well written, concise, and includes appropriate figures. I recommend publication after the following minor issues and suggestions are resolved or considered.*

We thank the reviewer for this overview and recommendation.

***2. Data and Methods*** *I think the order of the data description could be changed. In the subsection about the radiosonde data, there is already written about Aeolus measurement geometry, overflights and a special Aeolus setting. It would be easier for the reader to understand this if he/she has learned about Aeolus before.*

Agreed, the order of the sections has been updated.

***2.2. Aeolus:*** *Somewhere it should be mentioned that Aeolus is a polar orbiting satellite.*

Agreed, we have mentioned this in the text.

*Line 72: Please consider making two sentences out of this (The satellite's orbit is sun-synchronous with 15.6 orbits each day and a repeat cycle of 7 days. For the duration of the observing period in this study there is a close overpass to the site of the Singapore rawinsonde station between 22:55 and 23:00 UTC every Wednesday.)*

We agree and we have changed the text to match the reviewer's suggestion.

*Consider offering the reader a reference for more detailed information about the Aeolus data products and processing (e.g., Reitebuch et al., 2018 and Tan et al. 2008).*

We have added the suggested references to the manuscript, and have also included the L2B Algorithm Theoretical Basis Document (Rennie et al., 2020) as well for the reader's reference.

*Are there important changes between Baseline 11 and Baseline 14 that the reader should know about?*

45 It is not expected that there will be any major differences in these baselines that would affect our results here. However, we have added details of the Baseline 13 parameterisation which particularly improves Rayleigh cloudy winds, and the Baseline 14 correction which addresses a seasonal ascending/descending node bias. We only mention these details for information for the reader, and because we refer to both improvements later in the manuscript. We think it is
50 important to mention that different processing baselines are available for the Aeolus dataset, and that future users should note which version they are using.

*What are random error sources that could affect the data quality during the analyzed period?*

The main sources of random error that could affect the Aeolus data quality are (a) signal photon shot noise, which arises from fluctuations in the number of arriving photons from the expected
55 average and (b) unwanted electronic noise, which is caused by other electronic components in the detection chain of the instrument. Both act to decrease the signal-to-noise ratio (SNR) and/or broaden the spectral width of the backscattered signal. Wind errors for single-point Aeolus measurements are found to be typically around 5 to 9 ms$^{-1}$ in this study and in previous literature (e.g. Rennie et al. [2021], Martin et al. [2021] and Lux et al. [2022]), although note that aggregated
60 wind measurements from multiple points can be much more accurate. This information regarding sources of random error has been added to the manuscript.

*Please add a sentence as to why only Rayleigh winds are considered. In the upper troposphere in the tropics, I would expect good quality Mie winds at cloud top level or within thin clouds.*

We have added a sentence to explain that this was done to simplify the processing of the analysis
65 at the start of the study, and we have mentioned that including Mie winds in future studies could be useful for the reasons the reviewer has given.

**2.3. ERA5**

*Consider including the formula how to calculate ERA5 HLOS wind.*

We have added the formula as suggested by the reviewer.

70 *Please provide some information on the quality of the ERA5 data in the upper troposphere/lower stratosphere. NWP models typically have large uncertainties at these altitudes that could potentially affect the results(?).*

We have added a paragraph providing information on some of the relevant biases in ERA5 which may affect its representation of the QBO and/or comparisons made in the upper tropo-
75 sphere/lower stratosphere, and have justified the usage of ERA5 for our study. We have also quantified some of the uncertainty in reanalysis winds at these altitudes, citing Kawatani et al. [2016] which discusses uncertainties in tropical zonal winds in different atmospheric reanalyses (excluding ERA5), and citing Healy et al. [2020], which shows root-mean-square zonal wind differences between GNSS-RO and ERA5 of up to 5 ms$^{-1}$ at 30 hPa.

80 ### 3. Results

**3.1. Aeolus observations of the QBO disruption**

*Line 168: repeating information (see Line 129)*

Yes, we have removed this sentence and modified the sentence at Line 129.

**3.2. Validation against reanalysis and radiosondes**

85 *Line 208: I'm not totally convinced, that ERA5 stratospheric wind errors are that small (due to large model errors at these levels). E.g., the warm bias is mentioned later in the text, what makes this statement slightly doubtful.*

We agree with the reviewer that ERA5 stratospheric wind errors may be comparable to the HLOS error from Aeolus. We have corrected the text to clarify this, and we also include additional references to Kawatani et al. [2016] and Healy et al. [2020]. We suggest that much of the tropopause bias seen in Fig. 4b is a result of biases in ERA5, rather than solely being a consequence of biases in Aeolus. This is particularly likely since a bias is also seen between ERA5 and the radiosonde, as reviewer #3 points out.

*Line 235/236: Is there any conjecture about this height dependence of the Aeolus bias?*

We have expanded our discussion of the height dependence of the Aeolus bias to give two citations of validation papers which show a similar morphology to the bias structure found in our study. We also suggest that this height dependence is related to the vertical wind shear which maximises at the same altitude, with the following addition: "Given the high vertical wind shear at these altitudes, as noted by Houchi et al. [2010], it is likely that the apparent dipole in this bias is related to local wind shear effects. On the one hand, Aeolus, through its long horizontal accumulation of measurements which form each wind profile, has the capacity to capture localised wind shear that the radiosonde might miss. On the other hand, radiosondes provide a better vertical resolution and can accurately capture regions of high wind shear, but only at the location where the radiosonde is taking measurements."

**3.3 Equatorial waves during the QBO disruption**

*Please add a reference about the wave filtering method that is used here.*

We have added the following references to justify using the Gaussian band-pass filter in this study to highlight the Kelvin waves: Holton [1973], Polo et al. [2008], Roundy and Kiladis [2006], Blaauw and Žagar [2018].

*Figure 6: What are the thick horizontal gray lines (early January 2020 and around March 2020)? I can't find a description in the caption or the text.*

The gray lines are data gaps. The figure caption has been updated to add this information.

**Introduction/Discussion**

*Regarding the current state of Aeolus research consider adding the reference Martin et al., 2023 (https://doi.org/10.5194/wcd-4-249-2023) about the impact of assimilating Aeolus observations on QBO (with and without QBO RBS) and ENSO in 2020 in the Introduction or Discussion*

Yes, this is a relevant reference, particularly since it discusses the QBO 2020 RBS. We have added it to the discussion.

**Technical stuff**

*I believe the EGU guidelines say that there should be a space in between for the unit ms$^{-1}$.*

Thanks, this has been fixed in the text.

*Sometimes you use 2019/2020, sometimes 2019-2020 or 2019-20. Please make this consistent.*

We have changed all instances to 2019/2020 for consistency.

*Line 83: please change 2,000 m –> 2.000 m*

This value is meant to be 2000 m rather than 2 m, so we have left this unchanged.

*Line 84: please write out the abbreviation UTLS once*

We have added this to the text.

*Line 105: please write out the abbreviation ECMWF once*

We have added this to the text.

*Line 155: please write out the abbreviation GNSS-RO once*

We have added this to the text.

*Line 272: please write out the abbreviation OLR once*

We have added this to the text.

*Line 358: abbreviation DWL is already defined in Line 174*

135    Yes, we have removed this from the text.

**References**

M. Blaauw and N. Žagar. Multivariate analysis of kelvin wave seasonal variability in ecmwf l91 analyses. *Atmospheric Chemistry and Physics*, 18(11):8313–8330, 2018. doi: 10.5194/acp-18-8313-2018.

140    S. B. Healy, I. Polichtchouk, and A. Horányi. Monthly and zonally averaged zonal wind information in the equatorial stratosphere provided by gnss radio occultation. *Quarterly Journal of the Royal Meteorological Society*, 146(732):3612–3621, 2020. doi: 10.1002/qj.3870.

J. R. Holton. On the frequency distribution of atmospheric kelvin waves. *Journal of Atmospheric Sciences*, 30(3):499–501, 1973. doi: 10.1175/1520-0469(1973)030<0499:OTFDOA>2.0.CO;2.

K. Houchi, A. Stoffelen, G. J. Marseille, and J. De Kloe. Comparison of wind and wind shear climatologies derived from high-resolution radiosondes and the ecmwf model. *Journal of Geophysical Research: Atmospheres*, 115(D22), 2010. doi: 10.1029/2009JD013196.

Y. Kawatani, K. Hamilton, K. Miyazaki, M. Fujiwara, and J.A. Anstey. Representation of the tropical stratospheric zonal wind in global atmospheric reanalyses. *Atmospheric Chemistry and Physics*, 16(11):6681–6699, 2016. doi: 10.5194/acp-16-6681-2016.

O. Lux, B. Witschas, A. Geiß, C. Lemmerz, F. Weiler, U. Marksteiner, S. Rahm, A. Schäfler, and O. Reitebuch. Quality control and error assessment of the aeolus l2b wind results from the joint aeolus tropical atlantic campaign. *Atmospheric Measurement Techniques*, 15(21):6467–6488, 2022. doi: 10.5194/amt-15-6467-2022.

A. Martin, M. Weissmann, O. Reitebuch, M. Rennie, A. Geiß, and A. Cress. Validation of aeolus winds using radiosonde observations and numerical weather prediction model equivalents. *Atmospheric Measurement Techniques*, 14(3):2167–2183, 2021. doi: 10.5194/amt-14-2167-2021.

160    I. Polo, A. Lazar, B. Rodriguez-Fonseca, and S. Arnault. Oceanic kelvin waves and tropical atlantic intraseasonal variability: 1. kelvin wave characterization. *Journal of Geophysical Research: Oceans*, 113(C7), 2008. doi: 10.1029/2007JC004495.

M. P. Rennie, L. Isaksen, F. Weiler, J. de Kloe, T. Kanitz, and O. Reitebuch. The impact of aeolus wind retrievals on ecmwf global weather forecasts. *Quarterly Journal of the Royal Meteorological Society*, 147(740):3555–3586, 2021. doi: 10.1002/qj.4142.

P. E. Roundy and G. N. Kiladis. Observed relationships between oceanic kelvin waves and atmospheric forcing. *Journal of Climate*, 19(20):5253–5272, 2006. doi: 10.1175/JCLI3893.1.

---

## Author Comment (AC2)

**Response to Reviewer 2**

**Overview**

We thank the reviewers for their helpful comments on our study. Based on their comments, the manuscript has been substantially improved since its initial submission. We briefly summarise the improvements here in an overview, before responding to the reviewer's specific comments.

The key improvements to the paper include, but are not limited to:

- A new section investigating equatorial waves, in response to comments by reviewers #2 and #3. This includes (a) new analyses of Kelvin wave variance during the QBO disruption (b) new analyses of the symmetric and antisymmetric power spectra, and (c) a consideration of the equivalent depth $h_e$ and vertical wavelength $L_z$ of Kelvin waves. These results help to clarify equatorial wave processes during the disruption, are consistent with our existing material and other studies, and have improved and expanded the scope of the our study.
- Statistical tests which have been applied to our results, including to the comparison of Aeolus with ERA5 and radiosonde measurements, and to our calculation of the equivalent depth.
- A more comprehensive description of important Aeolus and ERA5 biases which might influence the results in this study, with additional information on these for the reader's benefit.

We now respond to the specific comments by reviewer 2 below.

**Reviewer 2**

*In their manuscript "Aeolus wind lidar observations of the 2019/2020 Quasi-Biennial Oscillation disruption with comparison to radiosondes and reanalysis", the authors observe the evolution of the second ever Quasi-Biennial Oscillation (QBO) disruption in novel Aeolus satellite measurements. The observations are validated with radiosonde and reanalysis data. In addition, the reanalysis and satellite data are analyzed with respect to Kelvin wave occurrence before the disruption.*

*This study is an excellent show case for the use of novel Aeolus wind measurements for the investigation of dynamic phenomena in the upper troposphere / lower stratosphere (UTLS) region. It clearly demonstrates the capabilities and restrictions of this unprecedented dataset for atmospheric research. With this the article is in general of high scientific significance. Nevertheless, I see a major need for improvement with respect to the scope of the journal and the scientific questions answered by the manuscript.*

*In its current state the paper is more a validation of the new Aeolus dataset and with this more in the scope of Atmospheric Measurement Techniques. However, by applying some changes and extending the discussion the paper could help to address important scientific aspects of QBO research. This would make it very suitable to Atmospheric Chemistry and Physics.*

*In detail, the paper would strongly benefit from a more detailed discussion on what can be learnt from the differences between ERA5 and Aeolus for the generation of the disruption. Which processes are different (e.g. Kelvin wave activity) and what does this tell us about our current understanding of the physical processes behind?*

We thank the reviewer for this overview and for their recommendations.

**Specific comments**

*L27: Any reference to support this sentence? Maybe Smith et al. (2022)?*

Yes, we have added Smith et al. (2022) which supports this claim.

*L29: Also, here a reference would provide additional information to the reader, e.g. ESA (2020a)*

We have added the suggested reference to the paper.

*L36ff: This sentence is really long, contains a lot of information and is hard to read. Maybe better split into two or more sentences.*

Agreed. We have split this sentence into two sentences.

*L39: What is your study adding to the current knowledge / understanding of the QBO? What are the research questions you are trying to answer? For the reader it usually helps to briefly address the "why exactly this research" in the introduction to easier follow the manuscript.*

Our study contains the first direct wind measurements of the QBO using a space-borne DWL instrument. The additional scientific analyses suggested by this reviewer have helped to shed light on why forecast models may have struggled to predict the disruption, which we have implemented by finding key differences in the representation of Kelvin waves between Aeolus and ERA5. We have added some text to the introduction to address the reasons for this research, as suggested.

*L46f: Global and local are antonyms, but how does reanalysis fit into the picture? Is a reanalysis perspective really a measurement perspective? I would suggest to reformulate this sentence.*

We have reformulated the sentence as suggested by the reviewer. The renanalysis of course gives a global perspective, so we've removed the inference that it is neither global nor local.

*Section 2.1: Maybe it is worth mentioning that the Singapore Radiosonde Station is commonly used as the gold standard when it comes to QBO analysis as it provides the longest available data record in the tropical stratosphere. Okay, this is somewhere later in the paper. Maybe you could move it forward to the dataset description.*

Agreed, we have moved this sentence into the dataset description as suggested by the reviewer.

*L66ff: Is Banyard et al. (2021) really an appropriate reference here? Maybe remove this part of the sentence. The references before already support this sentence sufficiently.*

This sentence is a close-quotation from Banyard et al. (2021) since this describes what we intend to say well and we do not wish to rephrase this, so a citation is necessary here to avoid plagiarism.

*L71: I would suggest to better reference ESA (2020b) here.*

We have added the reference ESA (2020b), although we have left the original reference of Legras (2022) in place since we did use this resource to help give us information about the Hunga-Tonga RBS.

*L77f: This is in general correct, but the sentence is confusing here. 35° off-nadir (as mentioned in the sentence before) per se gives mainly the vertical wind component. Only because the vertical wind component is much smaller than the horizontal component, the zonal wind can be derived, which you actually mention, but only one sentence afterwards. So, the order of the sentences here confused me. Maybe just remove this sentence on the advantage for QBO observation?*

Yes. We have rephrased this paragraph to avoid confusion.

*L108: ERA5 is available at a temporal resolution of 1 hour:*
*https://www.ecmwf.int/en/forecasts/dataset/ecmwf-reanalysis-v5*

We used 3 hour resolution data in order to reduce unnecessary computational load. In practice such a change makes only an extremely small difference to our dataset and does not affect our results significantly.

*Figure 1: How did you fill the data gaps in this plot? E.g. between the weekly QBO settings or during instrument down times (e.g. spring 2021).*

A 7-day boxcar filter is used to fill the gaps above 20 km following the QBO 2020 RBS change,

and where there are data gaps elsewhere, these are filled using a broader 20-day boxcar filter. We have added this text to the figure caption. As this zonal-mean timeseries requires less stringent constraints for accuracy, we have been able to apply the intended restrictions for NWP centers with some flexibility.

*L116f: This would probably be better visible if the Aeolus dataset would be extended until launch in 2018.*

We have only used data from after the switch from laser FM-A to FM-B occurred in June 2019, since at the time this analysis was conducted, data from the first FM-A data period was not trustworthy. Since then however, data from this period has been reprocessed to remove the existing biases, so if this analysis was done in future we would indeed extend back until launch in 2018 as the reviewer suggests.

*L161: Why only for reanalysis?*

We have added 'and NWP models' to the text.

*L185: ..., as \*mentioned / described\* on Kawatani et al. (2016), ...*

We have changed the text as suggested.

*L185ff: I had to read these sentences a couple of times, before understanding their meaning. Maybe rephrase?*

Agreed, we have rephrased this paragraph to make it clearer, and moved the sentence about the importance of using the Singapore radiosonde into the Data and Methods section.

*L193: Why 3D?*

Removed to avoid confusion.

*L204f: This is a courageous statement. Weiler et al. (2021) and Abdalla et al. (2020) only look into this on global average. According to ESA (2021), there could be higher regional biases. In addition, there are small bias differences between ascending and descending orbits in certain months (also ESA, 2021).*

We agree with the reviewer and have added a distinction between global and regional biases in the text. We have also modified the text to change "kept to a minimum" to "kept low".

*L206: I fully agree to use negative HLOS in figure 3, but please also state this in the figure or caption.*

We have clarified this in the figure caption.

*L211ff: You nicely describe the evolution of easterly winds in the profiles. The weakening of westerly winds is also nicely visible in profiles b-e. Maybe you could also guide the reader through this point first, before coming to the evolution of the easterlies. Otherwise, why do you show the profiles b-e?*

Yes, we agree that this is important. We have added a sentence to guide the reader through the weakening of westerly winds which occurs before the disruption easterlies develop, so that the full context of the disruption itself is provided.

*L219: Aeolus measurements are performed separately for each range bin and, unlike the derivation of temperature and aerosol backscatter and extinction, the derivation of wind from lidar measurements does not rely on an iterative profile reconstruction. Thus, cloud contamination in higher atmospheric bins cannot originate from clouds in lower atmospheric bins. To me it seems more that your limit of 250km is quite a large range (in validation studies often 100km is used as colocation criterion) and the atmosphere varies within this distance. There are many points in your profile comparisons where two or more Aeolus winds in the same range gate at the same time are quite far from each other (not only in figure 3g & 3h). This is probably due to the variability of the wind in latitudinal direction. Maybe the altitude offset in figure 3f and 3i are also due to a not perfect match in location.*

Firstly, we concur with the reviewer's assessment of the Aeolus measurement profile retrieval. Regarding the comment on cloud contamination issues, we speculate that the higher random error in the higher atmospheric bins might result from cloud contamination at those altitudes, rather than from the clouds at lower altitudes, which were removed through quality control. Aeolus' along-track accumulation spans approximately 87 km, and data may still be flagged as clear-sky even if some clouds are present. Considering the presence of convection at lower levels, isolated deep convection might also occur along this 87 km track, albeit insufficient to flag high-altitude data as cloud-contaminated.

Secondly, we agree that 250 km is quite a large range so we have redone the analysis for 150 km colocation. We found that this is a good trade off between sufficient data for our analysis and close colocation (The average minimum proximity of the Aeolus overpass to the radiosonde station is ~80 km). A similar radiosonde comparison by Ern et al. (2023), albeit implemented for a slightly different purpose, imposes colocation criteria of $\pm 2°$ latitude and $\pm 10°$ longitude which, although a larger area, should still provide a good comparison of the larger-scale dynamics. Interestingly we find that the altitude offset in figure 3f and 3i is still present in the 150 km colocation results, which suggests it may not be due to latitudinal variability. In addition, the 150 km colocation shows, if anything, a slightly weaker similarity to the radiosonde, especially in figure 3e, likely due to the reduced number of profile points used. Nevertheless, Fig. 3 and 4 have now been updated to constrain data to a maximum of 150 km from the Singapore radiosonde launch site.

*Figure 4: Which colocation method is used for the comparison of these data points? Aeolus vs radiosonde, probably the same 200km, but what did you do for the comparison to ERA5? Interpolate the model onto the Aeolus and radiosonde location? This might explain the better agreement between ERA5 and Aeolus vs radiosonde and Aeolus.*

We have used the same colocation method as for figure 3, now updated to contain all colocations less than 150 km. The profiles in figure 3 are a selected subset of all the profiles used for figure 4. The ERA5 data has been projected on the HLOS measurements as described in the Data and Methods section.

*L251ff: Maybe a good idea for a next study. ECMWF forecast / analysis data with and without Aeolus assimilation is available at ECMWF.*

We thank the reviewer for their good suggestion, this is indeed an idea for further study which the authors would be interested in carrying out.

*L255: What exactly do you mean with like-for-like comparison? Are you interpolating ERA5 data to Aeolus measurement locations?*

Yes, that's precisely what we do. Although other studies may compare data from Aeolus against ERA5 on different grids, since we're focusing on the perspective of Aeolus we have interpolated ERA5 onto each Aeolus measurement location. We have altered the text in the Data and Methods section to clarify this, and added the formula used to convert ERA5 $u$ and $v$ into a synthetic ERA5 HLOS wind. In the text here, we have removed the phrase 'like-for-like' to avoid confusion, and have referred the reader to the ERA5 subsection for clarity.

*Figure 5: Maybe you could show the ERA5 contour lines in the whole plot.*

We can only show ERA5 contours over the same area as the Aeolus measurements since we are effectively making a comparison between (a) Aeolus measuring the real atmosphere and (b) Aeolus measuring the 'ERA5 atmosphere'. This means that the 'ERA5' dataset has the same spatial and temporal extent as the Aeolus dataset.

*L257: You not only see this difference in onset time, you also clearly see stronger winds and wind gradients in the Aeolus data in the troposphere before the disruption (e.g. -5 m/s line is at a higher altitude in Aeolus data). These stronger wind gradients might have an impact on the Kelvin wave propagation (you discuss afterwards), so I think they should be mentioned here.*

We thank the reviewer for this useful comment. It has partially motivated some of the further analysis added later in the paper. We agree, and the updated version of figure 5 shows this more clearly. We have added more information about the relationship between these stronger wind gradients and Kelvin wave propagation to the text.

*L270: ... highlight only symmetric wind structures ... (antisymmetric waves with respect to the equator are removed due to your averaging from -5° to +5° latitude; for a detailed analysis of equatorial waves in the Aeolus dataset, you could have a look at Ern et al. 2023)*

That is correct. We have added this clarification.

*L291 – 296: This paragraph mixes different things and draws conclusion which I either do not understand correctly or a very far-fetched. I would suggest to remove the whole paragraph.*

We have removed most of this paragraph, and only retained the sentence relating to the comparison between the different QBO disruptions, since we believe that this is particularly relevant to this study.

*animation S2: Title of plot and caption are not in line with each other. It should be +-5° latitude as everywhere in the manuscript.*

We thank the reviewer for pointing this out. The Hovmoller and cross-section plots are plotted from the equatorial bin of a 5° latitude resolution gridded dataset, so these should both be +-2.5 degrees. A gridded dataset is used to make the analysis easier to run. The zonal-mean time series of the QBO for figure 1 and 5 is for +-5° since this Aeolus data has not been binned in the same way; in order to maximise the SNR and accuracy as much as possible. Re-running code to ensure these match would take a long time and is unlikely to change the timeseries figures significantly.

*Animation S2 and Figure 7: Why don't you apply the same temporal filtering as in the Hovmöller plots? You want to show the Kelvin waves, but these are barely visible due to the dominating feature of the Walker circulation. By applying a similar filter as in Figure 6, this strong dipole should vanish and the Kelvin waves should become clearly visible.*

We agree with the reviewer here and have updated animation S2 and figure 7 to apply the same temporal filtering as in the Hovmöller plots as suggested. The result of this is that the Walker circulation no longer dominates and the Kelvin waves have become more clearly visible.

*L328ff: You could perhaps reformulate the sentence to stress the importance of a future wind lidar measuring up to at least 30km or higher for QBO research.*

We have added this to the text and given two references in support of this.

*L336: ... this change in random error ... (the high random error itself is a problem especially for short analysis periods and perturbation analysis, for these a bias would be less of a problem)*

We have edited the text to add this, although this paragraph has now been moved to the Data and Methods section.

*L348: Why only reanalysis model?*

Yes, this is the case for NWP models as well. We have added this to the text.

*L349: Why is your analysis not spanning the whole Aeolus measurement period, so why is the data before summer 2019 missing?*

As mentioned earlier, we have only used data from after the switch from laser FM-A to FM-B occurred in June 2019, since at the time this analysis was conducted, data from the first FM-A data period was not trustworthy. Additionally, our analysis does not span the whole Aeolus measurement period for a very important reason, which is that we are focused primarily on the QBO disruption of 2019/2020. It is important that the focus of the paper does not shift away from the disruption (and therefore relevant for publication in ACP) towards a more general techniques paper on the characteristics of Aeolus measurements in the tropical UTLS (which would be more

relevant in AMT). Please see comments by reviewer #3 for more details.

*Discussion: This is not a general discussion of the results of the study. What you describe here are drawbacks of the Aeolus mission. Thus, I would suggest to either rename the section or revise its content.*

We agree with the reviewer that there is an overemphasis on the drawbacks of the Aeolus mission here, and so we have revised the contents of the discussion section accordingly. We think it is important to still mention key limitations of the study which may affect the results, but some of the content has been moved to the Data and Methods section, partially in response to a comment by reviewer #1 as well.

*L379f: Why is this in agreement with the data validation?*

We agree that this was confusing, and since we discuss the data validation in the previous paragraph, we have removed this clause for clarity.

*L385: Maybe better: ... have been discussed.*

We have changed this as suggested.

*L389: Maybe good to stress here the importance of measurements up to at least 30km.*

We have added this to the Discussion in response to an earlier comment by this reviewer.

*Conclusions: You nicely describe the data, but what have we learned from a scientific point of view (except that Aeolus is a well-suited dataset for observing wind related phenomena in the UTLS)? Why did it come to this QBO disruption? Where is the difference between ERA5 and Aeolus and what can we learn from this difference for our understanding of the underlying physical processes? These are questions I would expect to be answered in an ACP manuscript.*

We thank the reviewer for this comment, which has motivated much of the additional scientific analysis we have conducted. We have learned that there are key differences in the vertical wind shear and Kelvin wave variances between ERA5 and Aeolus which help to explain the observed lag in disruption onset in the reanalysis. We have therefore made the manuscript more suitable for publication in ACP due to the greater focus on the dynamics of the QBO disruption.

---

## Author Comment (AC3)

**Response to Reviewer 3**

**Overview**

We thank the reviewers for their helpful comments on our study. Based on their comments, the manuscript has been substantially improved since its initial submission. We briefly summarise the improvements here in an overview, before responding to the reviewer's specific comments.

The key improvements to the paper include, but are not limited to:

- A new section investigating equatorial waves, in response to comments by reviewers #2 and #3. This includes (a) new analyses of Kelvin wave variance during the QBO disruption (b) new analyses of the symmetric and antisymmetric power spectra, and (c) a consideration of the equivalent depth $h_e$ and vertical wavelength $L_z$ of Kelvin waves. These results help to clarify equatorial wave processes during the disruption, are consistent with our existing material and other studies, and have improved and expanded the scope of the our study.
- Statistical tests which have been applied to our results, including to the comparison of Aeolus with ERA5 and radiosonde measurements, and to our calculation of the equivalent depth.
- A more comprehensive description of important Aeolus and ERA5 biases which might influence the results in this study, with additional information on these for the reader's benefit.

We now respond to the specific comments by reviewer 3 below.

**Reviewer 3**

*The manuscript by Banyard et al. presents wind measurements of the 2019-2020 QBO disruption from the first spaceborne Doppler wind lidar ADM-Aeolus. The general topic is relevant for publication in ACP and the paper is interesting, demonstrating that the QBO is generally well captured in Aeolus data. However, I find the manuscript very focused. It would benefit from including a broader analysis and discussion of the new information brought by Aeolus on tropical lower stratospheric dynamics compared to a state-of-the-art reanalysis. Some differences between the ERA5 and Aeolus are already mentioned but not much commented.*

*For these reasons, major revisions (additions) are required before the paper can be considered for publication. My recommendation to the authors would be to make more quantitative statements and expand the comparison between Aeolus and ERA5 to other modes of variability than the QBO (e.g., equatorial waves) which are only alluded to in the present manuscript. The authors should also consider the recent preprint by Ern et al. (ACPD) on a related topic.*

We thank the reviewer for their comments, and in response we have made significant changes to make the paper more quantitative and have expanded the comparisons, as described in the Overview at the top. We have considered and referenced the recent preprint by Ern et al. (ACPD) where appropriate.

**Main comments**

*1) Lack of significance test : in a few instances the author claim that a bias/difference is significant, but do not include a statistical significance test. For instance the near-tropopause wind bias at Singapore (l238-240) is not clear to me, since it is also present in the ERA5-Singapore radiosonde comparison.*

The authors agree that a statistical test is useful to quantify, for example, the near-tropopause wind bias at Singapore (L238-240), and we have included a Student's t-test of p<0.001 to highlight altitudes where the bias is statistically significant. A t-test is suitable here since the distributions are near-normal (as shown in the violin plots). As mentioned in the response to reviewer

#1's comment at L208, we suggest that much of the tropopause bias seen in Fig. 4b is a result of biases in ERA5, rather than solely being a consequence of biases in Aeolus. This is particularly likely since a bias is also seen between ERA5 and the radiosonde, as reviewer #3 mentions here.

*2) The authors touch the topic of equatorial wave representation in ERA5, but they could make more quantitative statements. They might also wish to change the colormap used for Figure*
*6. Given that one of the advantages of Aeolus is global sampling, the authors could without much effort quantify the differences as a function of longitude, beyond the location of Singapore. This is of interest, in particular since earlier studies (e.g., Baker et al., 2013 ; Podglajen et al., 2014) found that reanalysis uncertainties and wind errors in the tropical UTLS were larger in regions without assimilated radiosonde observations, resulting in a pronounced zonal structure.*
*Note that the need for wind observations to better constrain the flow in tropical regions is what motivated Aeolus in the first place.*

The authors agree that the topic of equatorial wave representation should be explored further, especially in the context of the disruption and with more quantitative statements. To this end, a new section has been added examining the differences in equatorial wave spectra between Aeo-
lus and ERA5 and differences in Kelvin wave equivalent depths with height. We have considered the reviewer's suggestion of further analysis of the differences between Aeolus and ERA5 as a function of longitude; however the recent preprint by Ern et al. (2023, ACPD) covers this longitudinal variation extensively and we have thus taken a different path, whereby we focus on the QBO disruption and Aeolus' capability to capture it and the QBO. The colormap for the middle
panel of figure 6 has been changed to make the structure clearer.

*3) Regarding the last sentence of the abstract : "This analysis highlights how Aeolus and future Doppler wind lidar satellites can deepen our understanding of the QBO, its disruptions, and the tropical upper-troposphere lower-stratosphere region more generally." , it is not shown in the paper that Aeolus can help clarify the mechanisms of the disruption, at least directly. I imagine*
*spaceborne lidars would help deepen the understanding of the QBO by putting observational constraints on equatorial waves and tropical gravity waves, but this is not really the focus of the paper.*

The authors have made changes to the paper to make it more suitable for publication in ACP by exploring the role of equatorial waves in the development of the disruption to a greater depth.
This better justifies this sentence in the abstract compared with previously, so we have kept it as in the original manuscript.

*Line 1 : The abstract could mention some key number (bias, etc.)*

We have added this information to the abstract, along with the key results from the new material.

*Line 102 : please describe the method briefly here*

We have moved these details a little earlier in the section and modified the text so that it describes the correct method.

*Line 238-240 : Is this a significant bias ? It does seem smaller than the difference between Aeolus and the radiosonde (panel a), which by the way, also maximizes near the tropopause.*

As we have previously mentioned, we have performed a Student's t-test on these results to
demonstrate that the bias is statistically significant. This is the case both for the difference between Aeolus and the radiosonde (panel a) and the differences between the observing instruments and reanalysis (panel b and c). Though significant, this does not give any direct indication as to the cause of the biases in relation to each dataset, i.e. it is not possible to say that one bias is the addition of the other two since the distributions are not additive.

*Line 264-265 it is suggested that the sampling by aeolus does not induce much bias. You could easily prove this with a figure comparing ERA5 with aeolus sampling and the actual zonal mean*

The biases in ERA5 that are caused by regional differences in observation sample sizes are discussed by Kawatani et al. [2016] and are mentioned earlier in our study. Given that Aeolus'

data sampling is mostly homogeneous (aside from data gaps due to instrument downtime or quality controls), we believe it is fair to make this assumption. We have not added the suggested figure since we do not think it adds much strength to the paper. Mapping the locations and density of input observations for ERA5 would take quite a bit of work to complete, and is outside the scope of this study.

*Fig 4 : the pressure scale is wrong in this figure. What do the dots correspond to? Some of the*
*numbers in the legend would find their place in the main text.*

The pressure scale has been corrected and the dots have now been referenced in the figure caption. We are not sure which legend the reviewer is referring to, but we have placed all relevant numbers off the figure in the caption. Median differences and standard deviations have been updated to correspond to the new 150 km constraint.

*Fig 5 : Could you rather plot the mean wind in contour and the difference in colors ? This would be more quantitative. There seems to be a delay, does it hold for the later period or is this just a feature of the disruption?*

The authors agree with the suggestion by the reviewer and have changed the plot so that it shows the mean wind for Aeolus and ERA5 in contour and the difference between them in colour. In
general, Aeolus has stronger winds than ERA5, but the zero wind line does not always follow the line of zero difference.

*Fig. 6 : Is there a zonal-mean structure in the error (see main comment 2) ? Also, I do not understand the need for time filtering. Eastward propagation appears clearly in your unfiltered plot.*

We disagree with the reviewer that time filtering is not necessary to more clearly show the eastward propagation of waves, since in the unfiltered plot the standing wave of the Walker Circulation dominates and masks the Kelvin waves. (There is also a contradictory comment from reviewer #2)

*lines 271-277 : The phrasing is a bit odd, suggesting that this long period was not an optimal*
*choice for Kelvin waves. You cite a few papers which show that the typical stratospheric Kelvin waves commonly seen in Hovmoller diagrams indeed have planetary wavenumber 1-2, periods of 10-20 days or more, as first described by Wallace and Kousky and reported in many papers and textbooks. Convectively-coupled Kelvin waves in the OLR are higher frequency but this is not what dominates at tropopause altitude where free-travelling waves are prominentl. I would*
*shorten this discussion.*

The reviewer makes an important point, and we have rephrased this discussion accordingly. The distinction between convectively-coupled Kelvin waves in the OLR (and therefore covering the large altitude range of 10-18 km as mentioned by Alexander et al. [2008]), travelling with periods of 7-10 days, and free-travelling waves in the TTL and stratosphere, with a larger range
of periods, is an important one to make. We have tried to keep the discussion concise whilst still mentioning the details we think are relevant. Our choice of filter is optimal for the dominant wave periods seen in the Aeolus data, and is broad enough to capture the vast majority of the Kelvin wave spectrum at these altitudes.

**References**

S. P. Alexander, T. Tsuda, Y. Kawatani, and M. Takahashi. Global distribution of atmospheric waves in the equatorial upper troposphere and lower stratosphere: Cosmic observations of wave mean flow interactions. *Journal of Geophysical Research: Atmospheres*, 113(D24), 2008. doi: 10.1029/2008JD010039.

Y. Kawatani, K. Hamilton, K. Miyazaki, M. Fujiwara, and J.A. Anstey. Representation of the
tropical stratospheric zonal wind in global atmospheric reanalyses. *Atmospheric Chemistry and Physics*, 16(11):6681–6699, 2016. doi: 10.5194/acp-16-6681-2016.

---

## Author Response (AR2)

**Response to Reviewers**

**Overview**

We thank the reviewers for their helpful comments on our revised manuscript. Both reviewers raise useful points and queries, all of which are addressed in detail individually below. Here, we provide a brief summary of the additional changes made, before we then respond to each reviewer's specific comments.

The key improvements to the paper include, but are not limited to:

- Added emphasis on the role of enhanced lower stratospheric westerlies in aiding the development of the QBO disruption.
- Clarification about the method for obtaining fig. 8 and the middle panels of fig. 9 and 10.
- An additional figure in the supplementary material to justify the removal of instrument noise for figures 9a, b, d and e.

We now respond to the specific comments by each reviewer below.

**Report #1 (Reviewer #3)**

*The content of the paper has been significantly expanded since the initial submission, and the study will be a valuable contribution to ACP. However, I have identified areas where additional clarification is needed, particularly regarding some of the new analyses and corresponding figures introduced in the revised manuscript. Furthermore, certain statements in the text could benefit from more substantial supporting analysis.*

We are grateful for the reviewer's support of our study as a valuable contribution to ACP, and are hopeful that the below clarifications and minor additions in response to their comments are sufficient for final publication.

**Major comments**

*1) Currently, it is not entirely clear how Figure 8 and the middle panels of Figures 9 and 10 are obtained. There are also inconsistencies between the text and figures (for instance, it is written that Fig. 8 depicts normalized perturbations but the colorbar indicates units of m/s). See specific comments for further details.*

We have modified the manuscript to clarify how the perturbations for figure 8 and the middle panels of figures 9 and 10 are obtained, and we apologise for the confusion about this. We have responded to this issue in more detail after the specific comment mentioned by the reviewer.

*2) The authors state in their conclusion that 'the larger values observed by Aeolus suggest that ERA5 does not capture as much breaking and dissipation of Kelvin waves with shorter vertical wavelengths, especially within the TTL. This leads to less westerly momentum being deposited between 18 and 21 km in ERA5, just above the region of greatest Kelvin wave variance as measured by Aeolus.' but this last point is not really supported by the current analysis, which does not include estimates of drag. Would it be possible to estimate momentum deposition by Kelvin waves, e.g. by adapting the approach employed by Alexander and Ortland (2010) to Aeolus observations ? This would strengthen the discussion.*

We acknowledge the reviewer's suggestion of including estimates of drag, and we agree that it would be useful to conduct such an analysis to confirm our hypothesis. As quoted by the reviewer, the strong suggestion made by our results is that ERA5 is depositing less westerly

momentum at these altitudes because it does not accurately capture the breaking and dissipation of Kelvin waves with shorter vertical wavelengths, especially in the TTL. Given that the purpose of our study was to characterise the QBO disruption using Aeolus, and that a study adapting the approach employed by Alexander and Ortland [2010] would be best done with a strong modelling component, this additional analysis is likely to be beyond the scope of this current study. We do however think this is an interesting piece of future work, and further modelling studies may be beneficial to investigate this drag more directly. We have also added a sentence to the discussion to reflect the reviewer's comment, as it certainly strengthens the manuscript's applications to future work.

*3) Figure 5 shows that the delay and underestimation of the QBO disruption in ERA5 compared with Aeolus is associated with weaker Westerlies around 18-19 km in the reanalysis. This is an important result, as recent studies (in particular, Kang et al., 2022), attribute a significant role to enhanced lower stratospheric Westerlies in the development of the disruption. I am under the impression that this point could be emphasized more.*

We agree with the reviewer that the delay in the QBO disruption's onset in ERA5 compared with Aeolus is an important result, particularly in light of other recent studies. To emphasise this point more as suggested, we have added: 1 sentence to the abstract; 1 sentence to the results section in the part where we describe figure 5; 2 sentences to the discussion.

**Other comments**

*Line 130: Consider referencing Bley et al. (2022) here, as this paper provides an estimate of Aeolus accuracy in the UTLS by comparing it with Loon long-duration balloon measurements.*

We have added a sentence to reference Bley et al. (2022) as suggested by the reviewer.

*Lines 156-157: It would be worth clarifying whether you apply a smoothing to the radiosonde data for the point-by-point comparison (or not).*

We have applied linear interpolation to fill gaps in the radiosonde data but we have not applied any additional smoothing. We have now clarified this in the text as suggested.

*Line 161: Similar question for ERA5. How do you address the discrepancies in vertical resolutions between the data sets?*

This is addressed by our creation of the 'synthetic ERA5 HLOS wind' data set, which is geometrically identical to the Aeolus data set, but instead simulates Aeolus orbiting through an ERA5-like atmosphere rather than the real one. The result is that the discrepancy in vertical resolution between Aeolus and ERA5 data is eliminated by our method.

*Line 232: When mentioning "fine vertical resolution", it might be helpful to reiterate the ∼1 km resolution for clarity.*

We have added "(∼1 km)" to the text here.

*Figure 3: Why are there several data points at the same altitude for Aeolus?*

There are sometimes several data points at the same altitude for Aeolus because multiple profiles often fall within the 150 km colocation radius, centred on the Singapore radiosonde launch site. The solid lines demarcate the average (mean) wind speed within this 150 km radius circle.

*Line 315-320 (also regarding the caption of Figure 4): I am wondering about the number of significant digits you use for the differences. I noticed a change in the values between the initial and current version of the manuscript, which (I imagine) comes from switching from 250 to 150 km collocation radius. This suggests to me that 2 significant digits is 1 too many.*

Although the shift in collocation radius from 250 to 150 km produces a slight change in values between the manuscript versions, we do not believe the difference is sufficient to justify limiting

the quoted values to 1 significant digit. The mean change is only 0.05 ms$^{-1}$, ranging from 0.02 to 0.07 ms$^{-1}$, and it is likely that the standard error on each value (which is the best way to determine how a result should be quoted, as described in Hughes and Hase [2010]), is less than 0.1 ms$^{-1}$. Given that we have reduced the collocation area by 64%, eliminating a rather large area both south of the equator and north of 2.5°N (latitude naturally being more important than longitude here), that the changes are this small in magnitude gives us confidence in our results. This is in spite of the relatively low reduction in the number of Aeolus profiles used. We also want to maintain consistency in the way we quote the different values here, as well as with other literature.

*Figure 5 : Very interesting!*

Yes, the changes made to this figure for the revised manuscript reveals very interesting differences between Aeolus and ERA5!

*Line 377-378: You could assess a potential regional bias, e.g. by showing altitude-longitude section of the average HLOS wind difference between ERA5 and Aeolus during the disruption (e.g. Jan-March). This is not done in Ern et al. (2023), who restrict themselves to radiosonde locations.*

Whilst we agree with the reviewer that potential regional biases during the disruption could play an important role in modulating both the time-lag of its onset in ERA5 compared with Aeolus, as well as the disruption's poor predictability in NWP, we have decided to leave this for a future study. Major comment 2 suggested a deeper analysis of the momentum deposition by Kelvin waves, and so we think a study which combines this more general view of differences in Kelvin wave propagation between models and Aeolus, with a closer analysis of the regional variation in these differences, could be a useful piece of future work. It is possible, for instance, that shorter wavelength Kelvin waves are captured less well by existing observations over the Pacific Ocean, where their propagation is also affected by the Pacific Walker Circulation. There are many other interesting questions of this nature which could be explored, but we feel that the addition of this new material to our particular study here could be detrimental to the overall focus on the 2019/2020 QBO disruption. Further suggestions of this nature are however very much welcome.

*Line 460: What do you mean by 'Kelvin wave-filtered'? Is it the same as the time filter introduced around line 390. If yes, you might consider replacing by 'time-filtered'. 'Kelvin wave-filtered' suggests to me that both time and longitude filters are applied together to select only positive phase speeds.*

Yes, this is the same time filter introduced at line 390. We agree that this could cause a misunderstanding and have replaced 'Kelvin wave-filtered' with 'time-filtered' for the reason the reviewer has given.

*Figure 8: See major comment 1. I am not sure what is shown in panel a). What is this median composite ? From both Fig. 6 and Fig. 7, I would expect much smaller median values over the period (given that we consider maximum instantaneous values around 15 m/s of a field to which a 5-25 day filter has been applied). Also, line 461-462, you write that you are "normalizing by the median RMS of the entire domain". Then the plotted field should dimensionless, but the figure indicates m/s for the unit. This needs to be clarified.*

Figure 8a shows two plots on a longitude-altitude cross-section along the equator. (1) In colour, at each longitude and altitude, are the median time-filtered zonal wind perturbations, after scaling (multiplying) by the ratio of (A) the median of the root-mean-square (RMS) time-filtered zonal wind perturbations to (B) the RMS of the median time-filtered zonal wind perturbations (scale-factor = A/B). This is shown to provide an accurate composite representation of the typical true wind speeds and vertical wind shear associated with a Kelvin wave in the real atmosphere during this time period. (2) In contours, is the vertical wind shear which corresponds to plot (1). The reviewer is correct that the median values alone are much smaller (roughly a factor of 10), so this scaling (previously called normalisation incorrectly, as although it brings the data to a "normal"

reference point, it does not cause the data to become dimensionless and between 0 and 1) is required to provide true values of the vertical wind shear. To clarify, the median is taken over the time dimension whereas the mean for the RMS is always taken over the domain, across altitude and longitude. Changes have been made to the manuscript to clarify what is shown in these figures.

*Line 537: 'not shown': it would be useful to include this figure in the supplement.*

Yes, we agree. We have added this figure to the supplementary material to justify the removal of instrument noise for figures 9a, b, d and e.

*Line 540 and Figure 9: Do I understand correctly that you are subtracting 0.013m/s in panel a and 0.016 m/s in panel d at each omega and k? This seems unnecessary to me and makes the method description a bit hard to follow. I would recommend showing the 'biased' spectra, starting the colorbar at 0 and revert the colormap (i.e., have white as its first color). With the current colormap, one reads that most of the spectrum saturates at the maximum value (since areas with amplitudes below the threshold are kept white).*

Yes, we believe it is important to remove this noise in the observation data set so that the important features of the spectra can be seen and compared between Aeolus and ERA5. We have experimented with some of the suggested changes to the colormap and colorbar range and found that the current figure remains the easiest to interpret, so we have left it as before with most of the spectrum set to be transparent (as opposed to white).

*Line 547-549: There seems to be an inconsistency between text and figure: the text specifies that spectra are scaled but, in the title of the colorbar, there is a multiplication symbol (instead of a division). The differences are on the order of a few percent, is that really significant ? What significance test are you using ? If it is a student t-test, an estimate of the standard deviation of the amplitude at each wavenumber/frequency is required, how is it obtained?*

We have checked and confirmed that both the text and the figure are correct, but we have made some changes to the text to reduce the risk of any misunderstanding. We are indeed scaling (multiplying) by the mean amplitude of the two data sets at each $k$ and $\omega$, and this is because we are most interested in the differences between Aeolus and ERA5 in the spectral regions which correspond to different types of equatorial wave. We have clarified that we are conducting a Student's t-test, and stippling on the figure shows where $p < 0.001$, limited to regions where $SNR > 1$ and scaled differences are greater than $0.1 \times 10^{-3}$. As the reviewer says, an estimate of the standard deviation of the amplitude at each wavenumber/frequency is required, and this is obtained from the $(\pm k, \pm 0.05\omega)$ window of surrounding points (typically 54 values in total), as described in the caption for figure 9. We have used a fairly stringent significance test here, so in the few places where stippling is observed, we have confidence that the null hypothesis can be rejected, with the exception of the regions exhibiting low wave amplitudes and hence lower percent differences as the reviewer suggests (especially in the range $-2 \leq k \leq 2$). We had already indicated this in the text, but have added further clarification.

*Line 562: Why not show the spectra up to the altitude range of the disruption?*

We found that the data sampling at 20 km and higher was not sufficient for a reliable spectral analysis using the method of Salby [1982] during this time period. A more complex data interpolation routine implemented over a longer time period may yield reliable spectra at these heights, however this is beyond the scope of this study.

*Line 594: I imagine that the linear regression is weighted by the amplitudes of the Kelvin waves. This information should be included.*

Yes, that is correct, we have added this information to the figure caption.

*Line 613: 'vertical wind information' should be replaced with 'horizontal wind vertical profile information'.*

185  Yes, thank you for spotting this error. We have now corrected this as suggested.

*Line 636: 'overlying', don't you mean 'underlying'?*

We were originally referring correctly to the overlying westerly winds here, however we have now edited the text to replace this with the equatorial waves beneath the easterly jet, since Aeolus doesn't really capture the overlying westerlies very well until the range-bin settings increase in
190  altitude in June 2020. This minor adjustment should now avoid any confusion for the reader.

*Line 642-643: I am missing part of the reasoning here. Do you want to imply that those propagating Kelvin waves partly dissipate and limit the magnitude of the Easterlies?*

Yes, the Kelvin waves appear to be propagating to and dissipating at a higher altitude in the reanalysis, which causes the easterly jet to be limited in magnitude. We have added a clause to
195  the text to clarify this.

*Typos and language:*

- Line 57: Replace "complimentary" with "complementary."
  Typo fixed.
- Line 102: Replace "each with thicknesses" with "each with a width" (singular).
200  Here, we have replaced "each with thicknesses" with "each with a thickness" to avoid confusion with the lateral width of either the range-bin or laser column.
- Lines 182-183: Adjust "wave, mean-flow interaction" to "wave-mean flow interaction."
  Adjusted as suggested.
- Line 261: Use "Figure 2" (following ACP guidelines, since it is the subject of the sentence).
205  Fixed.
- Line 261: Replace "nudges" with "constrains"
  Replaced as suggested.
- Line 515: Change "in the same way as" to "with the same implementation as."
  Changed as suggested.
210  - Line 627: Replace "showed" with "shown."
  Typo fixed.
- Line 630: This sentence does not contain a finite verb.
  We have replaced "A result that is in agreement with Bley et al. (2022)." with "This result is in agreement with Bley et al. (2022)." to fix this sentence.

**Report #2 (Reviewer #2)**

215

*The authors significantly improved the manuscript by extending it by a more in-depth analysis of the impact of equatorial waves on the QBO. I have only two small remarks / questions to this version of the manuscript. After these are clarified I recommend the publication in ACP.*

We thank the reviewer for their comments and for their recommendation of publication following
220  minor revisions.

*Line 487 (of manuscript with tracked changes): Do you observe Doppler shifting in your data or is this sentence more on the general theory of what you would expect to happen?*

We have added a clause to make it clear that this sentence is more on the general theory of what we would expect to happen, as the reviewer suggests. Our conclusions assume that any Doppler
225  shifting does not greatly affect the results, however studies such as Yang et al. [2012] did find that their wind power spectra fitted the theoretical dispersion curves better when taking this effect into account. They used ERA-Interim and ERA-40 winds for their analysis, and we would be quite interested to know if a similar analysis, conducted on either model data or a future reanalysis which assimilates Aeolus winds, could provide some interesting new results relating to this.

230  *Lines 609ff (of manuscript with tracked changes): ERA5 has at this altitude range a slightly*

*higher vertical resolution. The slightly higher vertical wavelengths observed by Aeolus might also be caused by aliasing to longer wavelengths due to the slightly coarser resolution. 6 km vertical wavelength is really at the edge of observability of Aeolus with a vertical sampling of 2 km.*

235 The reviewer makes an important point about the aliasing of shorter vertical wavelength Kelvin waves in Aeolus data. We are obtaining the vertical wavelength from our measurements of the equivalent depth at each height (3 km-deep bins), which is derived using the Salby [1982] method and therefore analyses only the temporal and longitudinal variations in wind. Using this method addresses some of the aliasing that might occur in Aeolus data for a single profile, or even a
240 single day. It is true that we are near the edge of observability for Aeolus, however most of the ERA5 winds at these altitudes are not constrained well by observations, so much of this finer structure is likely simulated by the reanalysis model, rather than coming from the assimilated observations. Our study therefore cautiously interprets these results, and emphasises consistency with earlier findings while refraining from overemphasising their magnitude due to Aeolus'
245 various observational limitations.

**References**

M. J. Alexander and D. A. Ortland. Equatorial waves in high resolution dynamics limb sounder (hirdls) data. *Journal of Geophysical Research: Atmospheres*, 115(D24), 2010. doi: 10.1029/2010JD014782.

250 I. Hughes and T. Hase. *Measurements and their uncertainties: a practical guide to modern error analysis*. OUP Oxford, 2010.

M. L. Salby. Sampling theory for asynoptic satellite observations. part i: Space-time spectra, resolution, and aliasing. *Journal of the Atmospheric Sciences*, 39(11):2577–2600, 1982. doi: 10.1175/1520-0469(1982)039<2577:STFASO>2.0.CO;2.

255 G-Y. Yang, B. Hoskins, and L. Gray. The influence of the qbo on the propagation of equatorial waves into the stratosphere. *Journal of the atmospheric sciences*, 69(10):2959–2982, 2012. doi: 10.1175/JAS-D-11-0342.1.